# A simulation study on the role of mitochondria-sarcoplasmic reticulum Ca²⁺ interaction in cardiomyocyte energetics during exercise

Ayako Takeuchi and Satoshi Matsuoka

*Department of Integrative and Systems Physiology, Faculty of Medical Sciences and Life Science Innovation Center, University of Fukui, Fukui, Japan*

Handling Editors: Vaughan Macefield & Yoshihiro Kubo

The peer review history is available in the Supporting Information section of this article (https://doi.org/10.1113/JP286054#support-information-section).

Mathematical integration of energetics

$$\frac{dATPi}{dt} = J_{ANT} \cdot Rmc + J_{AK} + J_{CK} - J_{ATPcons}$$

$$\frac{dNADH}{dt} = -J_{C1} + J_{PDHC} + J_{ICDH} + J_{OGDH} + J_{MDH}$$

⋮

The Journal of **Physiology**

**Abstract figure legend** An integrated model of human ventricular myocyte with excitation-contraction-energetics coupling was created to systematically analyse the contribution of Ca²⁺ regulation of mitochondrial enzymes to cardiomyocyte functions during workload transition; i.e. exercise. During exercise, the mitochondria-sarcoplasmic reticulum Ca²⁺ interaction, via uneven distributions of the mitochondrial Ca²⁺ uniporter protein MCU, Na⁺-Ca²⁺ exchanger protein NCLX, sarcoplasmic reticulum ryanodine receptor RyR and Ca²⁺ pump SERCA, optimizes the homeostasis of NADH, a key intermediate connecting mitochondrial metabolism and oxidative phosphorylation.

**Ayako Takeuchi** started a 'physiome' study – a combination of experiments and mathematical modelling – on cardiac cell volume regulation at the Kyoto University Graduate School of Medicine (Professor A. Noma's lab), Japan in 2004–2006. She carried out structure-function studies of Na⁺-K⁺ ATPase at the Rockefeller University (Professor D. Gadsby's lab) in 2007–2008. After coming back to Japan, she started a physiome study on mitochondrial Ca²⁺ dynamics to elucidate the physiological roles at the Kyoto University Graduate School of Medicine (2009–2013) and at the University of Fukui (2013–).

The Journal of Physiology

**Abstract** Previous studies demonstrated that the mitochondrial $Ca^{2+}$ uniporter MCU and the $Na^+$-$Ca^{2+}$ exchanger NCLX exist in proximity to the sarcoplasmic reticulum (SR) ryanodine receptor RyR and the $Ca^{2+}$ pump SERCA, respectively, creating a mitochondria-SR $Ca^{2+}$ interaction. However, the physiological relevance of the mitochondria-SR $Ca^{2+}$ interaction has remained unsolved. Furthermore, although mitochondrial $Ca^{2+}$ has been proposed to be an important factor regulating mitochondrial energy metabolism, by activating NADH-producing dehydrogenases, the contribution of the $Ca^{2+}$-dependent regulatory mechanisms to cellular functions under physiological conditions has been controversial. In this study, we constructed a new integrated model of human ventricular myocyte with excitation-contraction-energetics coupling and investigated systematically the contribution of mitochondria-SR $Ca^{2+}$ interaction, especially focusing on cardiac energetics during dynamic workload transitions in exercise. Simulation analyses revealed that the spatial coupling of mitochondria and SR, particularly via mitochondrial $Ca^{2+}$ uniport activity-RyR, was the primary determinant of mitochondrial $Ca^{2+}$ concentration, and that the $Ca^{2+}$-dependent regulatory mechanism facilitated mitochondrial NADH recovery during exercise and contributed to the stability of NADH in the workload transition by about 40%, while oxygen consumption rate and cytoplasmic ATP level were not influenced. We concluded that the mitochondria-SR $Ca^{2+}$ interaction, created via the uneven distribution of $Ca^{2+}$ handling proteins, optimizes the contribution of the mitochondrial $Ca^{2+}$-dependent regulatory mechanism to stabilizing NADH during exercise.

(Received 16 June 2024; accepted after revision 15 August 2024; first published online 10 October 2024)
**Corresponding author** A. Takeuchi: Department of Integrative and Systems Physiology, Faculty of Medical Sciences, University of Fukui, Fukui, 910-1193, Japan. Email: atakeuti@u-fukui.ac.jp

## Key points

- The mitochondrial $Ca^{2+}$ uniporter protein MCU and the $Na^+$-$Ca^{2+}$ exchanger protein NCLX are reported to exist in proximity to the sarcoplasmic reticulum (SR) ryanodine receptor RyR and the $Ca^{2+}$ pump SERCA, respectively, creating a mitochondria-SR $Ca^{2+}$ interaction in cardiomyocytes.
- Mitochondrial $Ca^{2+}$ ($Ca^{2+}_{mit}$) has been proposed to be an important factor regulating mitochondrial energy metabolism, by activating NADH-producing dehydrogenases.
- Here we constructed an integrated model of a human ventricular myocyte with excitation-contraction-energetics coupling and investigated the role of the mitochondria-SR $Ca^{2+}$ interaction in cardiac energetics during exercise.
- Simulation analyses revealed that the spatial coupling particularly via mitochondrial $Ca^{2+}$ uniport activity-RyR is the primary determinant of $Ca^{2+}_{mit}$ concentration, and that the activation of NADH-producing dehydrogenases by $Ca^{2+}_{mit}$ contributes to NADH stability during exercise.
- The mitochondria-SR $Ca^{2+}$ interaction optimizes the contribution of $Ca^{2+}_{mit}$ to the activation of NADH-producing dehydrogenases.

## Introduction

Mitochondrial $Ca^{2+}$ ($Ca^{2+}_{mit}$), which dynamically changes in response to changes in the cellular activity, is an important factor for cellular energy metabolism, apoptosis and cytoplasmic $Ca^{2+}$ ($Ca^{2+}_{cyt}$) signalling (see reviews by Brown et al., 2017; O'Rourke et al., 2021). In cardiomyocytes, this $Ca^{2+}_{mit}$ dynamics is determined by $Ca^{2+}$ influx via mitochondrial $Ca^{2+}$ uniport (CaUni) activity and $Ca^{2+}$ efflux via mitochondrial $Na^+$-$Ca^{2+}$ exchange ($NCX_{mit}$) and $H^+$-$Ca^{2+}$ exchange ($HCX_{mit}$) activities, with the former playing a dominant role (Carafoli et al., 1974; Nicholls & Crompton, 1980). In the past 15 years, the genes responsible for these activities have been identified – MCU complexed with accessory proteins such as MICUs, EMRE, and so on constituting CaUni activity, NCLX for $NCX_{mit}$ activity, and Letm1 or TMBIM5 for $HCX_{mit}$ activity – and their roles in cardiomyocyte functions have been extensively studied (Austin et al., 2022; Kwong et al., 2015; Luongo et al., 2015; Natarajan et al., 2020; Takeuchi et al., 2013; see also reviews by Boyman et al., 2013; Takeuchi & Matsuoka, 2021; Takeuchi et al., 2015; Wu et al., 2015).

Mitochondria are densely and regularly aligned in ventricular myocytes, and heterogenous localization of Ca$^{2+}_{mit}$ handling proteins and their contributions to spatial Ca$^{2+}$ movements inside the myocytes have attracted the attention of many researchers. De La Fuente et al. (2016) reported that in mouse and rat ventricular myocytes distributions of MCU and EMRE are biased towards the mitochondria-sarcoplasmic reticulum (SR) interface, which the SR Ca$^{2+}$ release channel ryanodine receptor (RyR) faces (De La Fuente et al., 2016). They defined the region as a 'Ca$^{2+}_{mit}$ uptake hotspot' and subsequently found that it lacks Ca$^{2+}_{mit}$ extrusion activity via NCLX (De La Fuente et al., 2018). Recently we independently found that NCLX and SR Ca$^{2+}$ pump (SERCA) were localized in close proximity to each other in mouse ventricular myocytes (Takeuchi & Matsuoka, 2022). We further demonstrated with mathematical model analysis that the experimentally recorded automaticity and Ca$^{2+}_{SR}$ dynamics of HL-1, a cell line derived from mouse atrial myocytes (Claycomb et al., 1998), could be reproduced only when spatial and functional couplings of mitochondria and SR via CaUni-RyR and NCX$_{mit}$-SERCA were assumed (Takeuchi & Matsuoka, 2022). These findings have filled the last gap in explaining the efficient Ca$^{2+}$ cycling between mitochondria and SR (De La Fuente et al., 2016, 2018). A question then arose: what are the physiological impacts of the spatial and functional couplings on the functions of intact ventricular myocytes, which have no automaticity?

Cellular energetics is one of the functions most likely to be affected by the above-mentioned couplings, because Ca$^{2+}$ is an important factor in both ATP consuming and producing processes. In cardiomyocytes, the ATP consuming process mainly occurs via myosin ATPase, SERCA, Na$^+$-K$^+$ ATPase ($I_{NaK}$) and the plasma membrane Ca$^{2+}$ pump ($I_{pCa}$), whereas ATP is produced mainly by oxidative phosphorylation in mitochondria (Katz, 2010). In order to adapt to the dynamic increase in cardiac workload, or ATP demand, during intense exercise, ATP supply increases rapidly as demonstrated by the linear relationship between workload and myocardial oxygen consumption (mVO$_2$) (Hata et al., 1994; Ingwall, 2002; Starling & Visscher, 1927). Feedback regulation by ATP hydrolysis products ADP and inorganic phosphate (Pi) has an important role for the adaptation, while allosteric regulation by Ca$^{2+}$ has also been proposed as an additional mechanism underlying the adaptation. This idea is based on an epoch-making finding that Ca$^{2+}$ activates mitochondrial dehydrogenases producing NADH, which is an intermediate product connecting mitochondrial metabolism and oxidative phosphorylation (McCormack et al., 1990; Rutter & Denton, 1988). Following controversies regarding the contribution of the Ca$^{2+}$-dependent regulations to cardiac energetics (see reviews by Aon & Cortassa, 2012; Beard & Kushmerick,

2009; Glancy & Balaban, 2012; Korzeniewski, 2017; Saks et al., 2006; Takeuchi & Matsuoka, 2020), we proposed the idea that the composition of metabolic substrates is an important factor determining the extent to which Ca$^{2+}$ contributes to mitochondrial energetics. Our simulation analysis using a detailed cardiac mitochondrial model demonstrated that, under the cytoplasmic substrate condition of malate/glutamate, which is frequently used in *in vitro* measurements of isolated mitochondria, steady state mitochondrial NADH (NADH$_{mit}$) concentration was elevated by increasing the Ca$^{2+}_{cyt}$ concentration. On the other hand, when other physiological metabolic substrates were assumed as the *in vivo* condition, Ca$^{2+}_{cyt}$ had marginal effects on the steady state NADH$_{mit}$ level (Saito et al., 2016), in line with experimental reports using isolated rat cardiac mitochondria (Vinnakota et al., 2011, Vinnakota, Singhal et al., 2016). However, it remains unclear how the Ca$^{2+}$-dependent regulatory mechanisms contribute to cardiac energetics and to overall cardiomyocyte function during dynamic workload transitions, such as exercise, when excitation-contraction coupling changes vigorously.

In the present study, we constructed a new integrated model of human ventricular myocyte with excitation-contraction-energetics coupling and investigated systematically the contribution of the mitochondria-SR Ca$^{2+}$ interaction to cardiac energetics as well as to overall cardiomyocyte functions during dynamic workload transitions in exercise. Simulation analyses revealed that the spatial coupling of mitochondria and SR, particularly via CaUni-RyR, is the primary determinant of Ca$^{2+}_{mit}$ concentration during exercise, and that it modulates the contribution of Ca$^{2+}_{mit}$-dependent regulation of NADH-producing dehydrogenases to the NADH$_{mit}$ dynamics.

## Methods

### Structure of the integrated human ventricular cell model with excitation-contraction-energetics coupling

The Integrated Human Ventricular Cell Model was created in Visual C# (Microsoft Visual Studio 2022) and the ordinary differential equations were integrated using the Runge-Kutta method with an adaptive time step in a range of $10^{-6}$ to $5.0 \times 10^{-3}$ ms. The source code is available on GitHub at https://github.com/atakeuti/IHVCM. The model consisted of a human ventricular myocyte model (Grandi et al., 2010), a contraction model (Shim et al., 2007), an interaction between mitochondria and SR (Takeuchi & Matsuoka, 2022), and a detailed mitochondrial energetics model (Saito et al., 2016) (Fig. 1*A*). All equations are presented

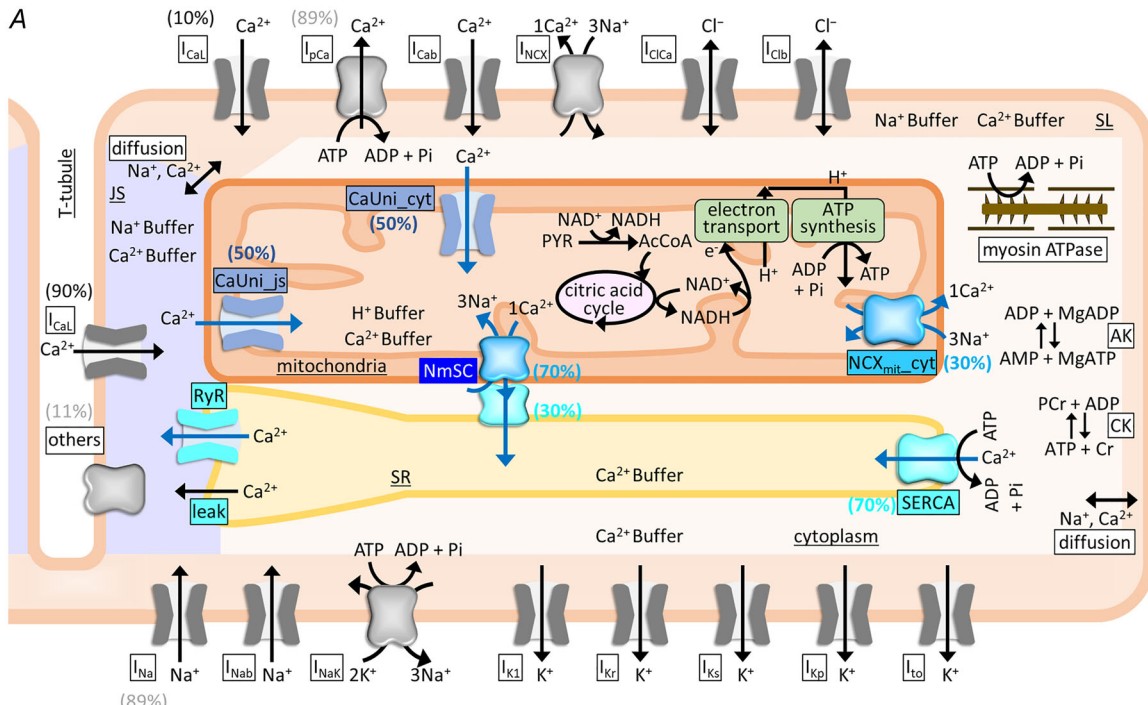

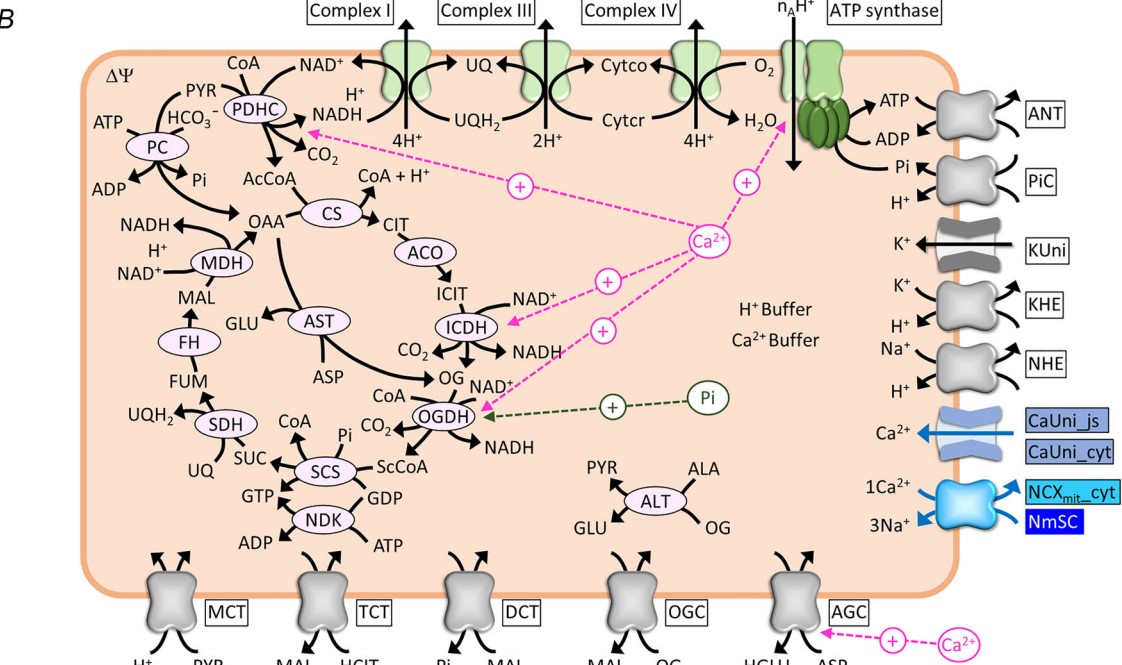

**Figure 1. Schematic presentation of the Integrated Human Ventricular Cell Model**

*A*) the overall view of the model, consisting of a human ventricular myocyte model, a contraction model, and a detailed mitochondrial energetics model, with an interaction between mitochondria and SR. Distribution of $I_{CaL}$ and other membrane channels/transporters (others) in JS/SL are indicated in parenthesis in black and grey, respectively. Uneven distribution of NmSC:NCX$_{mit\_}$cyt and NmSC:SERCA facing cytoplasm is set as, respectively, 70%:30% and 30%:70%. Uneven distribution of CaUni_js:CaUni_cyt is set as 50%:50%. *B*) the scheme of the mitochondrial energetics part of the model which is simplified in *A*. Allosteric regulations by $Ca^{2+}$ and Pi are indicated by magenta and green, respectively. AcCoA, acetyl CoA; ACO, aconitase; AK, adenylate kinase; ALA, alanine; ALT, alanine aminotransferase; ANT, adenine nucleotide translocase; ASP, aspartate; AST, aspartate

aminotransferase; Buff, buffer; CK, creatine kinase; Cytco, oxidized form of cytochrome c; Cytcr, reduced form of cytochrome c; DCT, dicarboxylate transporter; FH, fumarate hydratase; FUM, fumarate; GLU, glutamate; HCIT, citrate; $I_{Cab}$, background Ca²⁺ current; $I_{Clb}$, background Cl⁻ current; $I_{ClCa}$, Ca²⁺ activated Cl⁻ current; $I_{K1}$, inward rectifier K⁺ current; $I_{Kp}$, plateau K⁺ current; $I_{Kr}$, rapidly activating K⁺ current; $I_{Na}$, voltage-dependent Na⁺ current; $I_{Nab}$, background Na⁺ current; $I_{NCX}$, sarcolemmal Na⁺-Ca²⁺ exchange current; $I_{to}$, transient outward K⁺ current; KUni, K⁺ uniporter; MAL, malate; NDK, nucleoside diphosphate kinase; OAA, oxaloacetate; OG, 2-oxoglutarate; OGC, 2-oxoglutarate/malate carrier PC, pyruvate carboxylase; PYR, pyruvate; ScCoA, succinyl CoA; SCS, succinyl CoA synthase; SDH, succinate dehydrogenase; SUC, succinate; TCT, tricarboxylate transporter; UQ, ubiquinone; $UQH_2$, ubiquinol.

in the Online Supplementary Material. In order to obtain ion concentrations comparable to the original Grandi model after implementing the above components, several parameters were fine-tuned. The amplitude factors for SERCA, RyR, SR Ca²⁺ Leak and $I_{NaK}$ were set to 1.5, 1.5, 2.3 and 0.9 times of the original, respectively. The Ca²⁺ diffusion permeabilities for junctional space (JS)-subsarcolemmal space (SL) and cytoplasm-SL were set 1.1 times of the original. One of the cytoplasmic Ca²⁺ buffer systems in the Grandi model, troponin C (TnClow), was replaced with that used in the contraction model (Shim et al., 2007).

## Contraction model

We adapted the contraction model by Shim et al. (2007) with minor modifications, to the Integrated Human Ventricular Cell Model. In particular, the Ca²⁺ binding rate of troponin, $Y_1$ (ms⁻¹), was modified to better reproduce the Ca²⁺$_{cyt}$ concentration-force relationship measured using human myocardium (Gwathmey & Hajjar, 1990) (Fig. 2).

$$Y_1 = 12.8 \times \left(\left[Ca^{2+}\right]_{cyt}\right)^{0.81}.$$

The magnitude factor $B_{eff}$, model fit parameter $L_0$, and total troponin concentration were fine-tuned to obtain contraction comparable to the original model after this modification. See details in the Online Supplementary Material Table S7. The increase in half-sarcomere length (hsmL) is achieved by increasing the external load ($F_{ext}$) applied to the model cell.

## Mitochondria-SR interaction (MSI) model

Based on super-resolution imaging studies on the uneven distributions of Ca²⁺$_{mit}$ and Ca²⁺$_{SR}$ handling proteins in ventricular myocytes (De La Fuente et al., 2016; Takeuchi & Matsuoka, 2022), the mitochondria-SR interaction (MSI) component developed in the HL-1 cell model (Takeuchi & Matsuoka, 2022) was implemented in the Integrated Human Ventricular Cell Model. In short, we assumed the NCX$_{mit}$-SERCA complex (NmSC) and the CaUni facing JS (CaUni_js) (see Fig. 1A). In the control MSI model, the amplitude factors for NmSC and CaUni_js were set to 70% and 50% of total NCX$_{mit}$

and total CaUni, respectively. The remaining 30% and 50% of NCX$_{mit}$ and CaUni, defined as NCX$_{mit}$_cyt and CaUni_cyt, respectively, were set to face the cytoplasm. In addition, 30% of SERCA was replaced with NmSC and the remaining 70% was set to face the cytoplasm. Functional coupling of NCX$_{mit}$ and SERCA was expressed by assuming that Ca²⁺, extruded from the mitochondria via the NmSC, directly enters the SR. When MSI was not considered (non-MSI) in the model, the fractions of CaUni_js and NmSC were set to 0%, and those of CaUni_cyt and NCX$_{mit}$_cyt to 100% (Fig. 10). The amplitude factor for SERCA in the non-MSI model was set to 1.43 times, i.e. 100%/70%, that of the MSI model.

The CaUni model (Takeuchi & Matsuoka, 2022) was updated by implementing the term Ca²⁺$_{mit}$-dependent regulation, Ca$_{mit}$_reg, according to the experimental

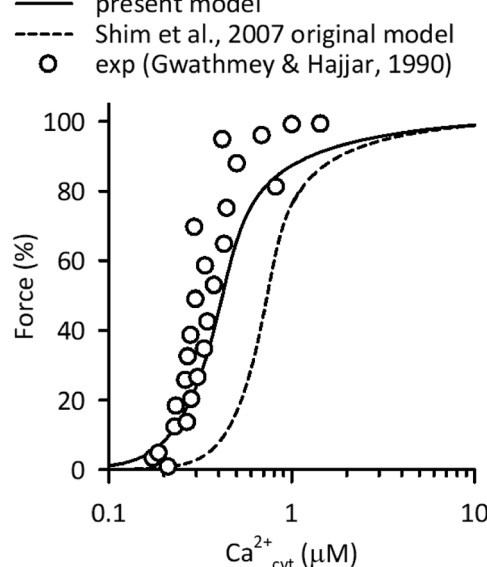

**Figure 2. Dependence of force in the contraction model on Ca²⁺$_{cyt}$**
The contraction model (Shim et al., 2007) with the present (continuous line) and original (dashed line) parameters applied, with $F_{ext}$ of 9.15 and 8.45, respectively, to obtain hsmL of 1.10 $\mu$m, under the condition of isometric contraction with various Ca²⁺$_{cyt}$ concentrations. The steady state force was obtained and was normalized to the maximal value. Experimental data using human cardiac muscle (exp; open circles) were from Gwathmey and Hajjar (1990).

report (Vais et al., 2016):

$$Ca_{mit\_reg} = 0.9 - \left( \frac{[Ca^{2+}]_{mit}}{[Ca^{2+}]_{mit} + K_{inh}} - 0.3 \right)$$
$$\times \left( \frac{K_{rec}{}^{n}}{[Ca^{2+}]_{mit}{}^{n} + K_{rec}{}^{n}} - 0.3 \right),$$

where the apparent inhibition constant of $Ca^{2+}_{mit}$, $K_{inh} = 5.0 \times 10^{-5}$ mM, the apparent recovery constant of $Ca^{2+}_{mit}$, $K_{rec} = 8.0 \times 10^{-4}$ mM, and the Hill coefficient $n = 2$.

The $NCX_{mit}$ model (Takeuchi & Matsuoka, 2022) was updated by increasing the $Ca^{2+}_{mit}$ affinity; i.e. reducing the apparent binding constant of $Ca^{2+}_{mit}$, $K_d Ca_{mit}$, from 0.0209 to 0.0025 mM, based on our electrophysiological measurements using mouse cardiac mitoplasts (Islam et al., 2020). Accordingly, the apparent binding constant of mitochondrial $Na^+$, $K_d Na_{mit}$, was reduced from 38.0000 to 18.7137 mM to satisfy the constraint of the equilibrium potential of $NCX_{mit}$, $E_{Na/Ca}$ (Kim & Matsuoka, 2008):

$$E_{Na/Ca} = 3E_{Na} - 2E_{Ca},$$

$$\frac{\left( K_d Na_{cyt} \right)^3 \times K_d Ca_{mit}}{\left( K_d Na_{mit} \right)^3 \times K_d Ca_{cyt}} = 1.0,$$

where $E_{Na}$, $E_{Ca}$, $K_d Na_{cyt}$ and $K_d Ca_{cyt}$ are the equilibrium potential of $Na^+$, equilibrium potential of $Ca^{2+}$, apparent binding constant of cytosolic $Na^+$ ($Na^+_{cyt}$), and apparent binding constant of $Ca^{2+}_{cyt}$, respectively.

The amplitude factors for CaUni and $NCX_{mit}$ were set to obtain comparable time courses of $Ca^{2+}_{mit}$ increase upon stimulus frequency change in rat trabeculae and guinea-pig ventricular myocytes (Brandes & Bers, 2002; Jo et al., 2006).

### Cardiac mitochondrial model

Our isolated mitochondrial model (Saito et al., 2016) was adapted for the energetics part of the Integrated Human Ventricular Cell Model. The model consisted of oxidative phosphorylation, the citric acid cycle, the pyruvate pathway, and substrate/ion transporters. $Ca^{2+}_{mit}$-dependent regulation of three dehydrogenases (pyruvate dehydrogenase (PDCH), isocitrate dehydrogenase (ICDH), and oxoglutarate dehydrogenase (OGDH)) and $F_1F_o$-ATPase (ATP synthase), and $Ca^{2+}_{cyt}$-dependent activation of aspartate/glutamate carrier (AGC) were implemented (Rutter & Denton, 1988; McCormack et al., 1990; Territo et al., 2000; Wescott et al., 2019; Fig. 1*B*). In the original Saito et al. (2016) model, we assumed that mitochondrial Pi ($Pi_{mit}$) allosterically activates Complex III of electron transport chain, based on the experimental report by Bose et al. (2003). Although this regulation was shown to contribute

to metabolic stability in various models including ours (Beard, 2005; Cortassa et al., 2006; Saito et al., 2016; Tran et al., 2015), it was subsequently denied experimentally (Bazil et al., 2016; Vinnakota, Bazil et al., 2016). Therefore, we decided to remove the $Pi_{mit}$-dependent activation of Complex III in the updated model.

In order to obtain resting $mVO_2$ comparable to that reported for the human hearts (e.g. 3.57 mM/min, calculated from $\sim$8 ml/100 g/min (Strauer, 1979)), the amplitude factors for mitochondrial trans-porters/channels/enzymes were doubled, except for the monocarboxylate transporter (MCT), which was multiplied by four to prevent depletion of mitochondrial pyruvate. The $mVO_2$ was 1.88 mM/min when stimulated with a cycle length of 1 s, on the same order of magnitude as the reported value, but slightly lower. This is possibly due to the difference between the estimates obtained by single cell model and by *in vivo* measurements.

Since the components for $Ca^{2+}$ fluxes, CaUni and $NCX_{mit}$, were replaced with those in the MSI model as described above, the $Ca^{2+}_{mit}$- and $Ca^{2+}_{cyt}$-dependent activities of ATP synthase and AGC, respectively, as well as the amplitude factors for $Na^+$-$H^+$ exchanger (NHE) and $H^+$ leak, were fine-tuned.

The updated isolated mitochondrial model well reproduced the *in vitro* experiments on $Ca^{2+}_{cyt}$-dependent changes of $NADH_{mit}$ and $mVO_2$ using isolated cardiac mitochondria (Appendix Fig. A*1A*) (Territo et al., 2000). In addition, qualitatively comparable results were obtained for cytoplasmic Pi ($Pi_{cyt}$)-dependency observed using isolated cardiac mitochondria (Bose et al., 2003) (Appendix Fig. A*1B*), albeit with deviations due to the removal of $Pi_{mit}$-dependent activation of Complex III.

### Basic characteristics of the integrated human ventricular cell model

The initial standard condition of the Integrated Human Ventricular Cell Model, which corresponds to the steady state condition for isotonic contraction stimulated with a cycle length of 1 s, is listed in the Online Supplementary Material Table S14.

Configurations of action potential, major ionic currents, $Ca^{2+}$ fluxes and $Ca^{2+}$ concentrations in each compartment are shown in the Appendix Figure A2. These are similar to the original Grandi model except for mitochondrial $Ca^{2+}$ dynamics, which was not implemented in the original model.

The frequency dependence of the model was examined in Fig. 3. The model was stimulated for 20 min at different frequencies under condition of isometric contraction with $F_{ext}$ of 4.25 to obtain 1.050 $\mu$m hsmL in similar manner to experiments of human ventricular trabeculae (Pieske

et al., 1995, 1999, 2002; Schwinger et al., 1993; Schmidt et al., 1998). The model well reproduced the experimental data on frequency-dependent changes in force and cytoplasmic ion concentrations (Fig. 3).

In the HL-1 cardiomyocyte, blocking $NCX_{mit}$ decreased $Ca^{2+}_{SR}$ concentration and decelerated $Ca^{2+}_{SR}$ reuptake in the caffeine application protocol (Takeuchi et al., 2013; Takeuchi & Matsuoka, 2022). Incorporation of the MSI component in the HL-1 model was necessary to reproduce these experimental results (Takeuchi & Matsuoka, 2022). In the Integrated Human Ventricular Cell Model considering MSI, $NCX_{mit}$ reduction decreased $Ca^{2+}_{SR}$ concentration and decelerated $Ca^{2+}_{SR}$ reuptake after caffeine application, whereas the reduction hardly affected them in the non-MSI model (Appendix Fig. A3), being comparable to those obtained using the HL-1 cell

model (Takeuchi & Matsuoka, 2022). Note that with the MSI model, $Ca^{2+}_{SR}$ concentration slightly increased when the amplitude factor for $NCX_{mit}$ was reduced from 1.0 to 0.5 and 0.3 (filled circles in Fig. A3*E*). Under these conditions, the $Ca^{2+}_{mit}$ concentration increased to several micromolar, and the residual NmSC was able to supply more $Ca^{2+}$ to the SR.

## Simulation protocols for dynamic workload change

To simulate dynamic workload changes in exercise, the following three interventions were applied to the model in different combinations; (1) shortening of stimulation cycle length mimicking heart rate increase, (2) increase in $F_{ext}$ mimicking ventricular end-diastolic volume increase,

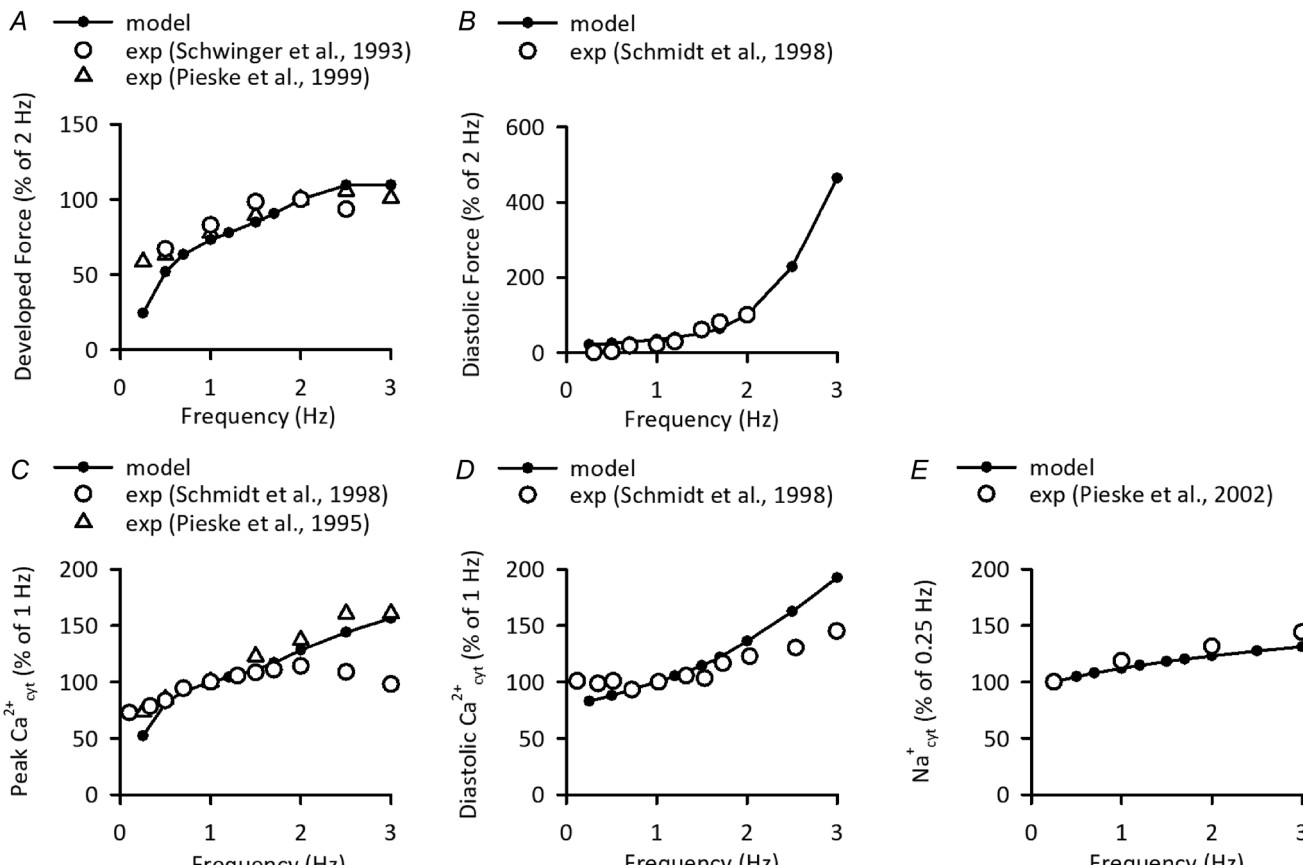

**Figure 3. Dependences of force and ion concentrations in the Integrated Human Ventricular Cell Model on stimulus frequency**
The Integrated Human Ventricular Cell Model was applied with $F_{ext}$ of 4.25 to obtain 1.050 $\mu$m hsmL under the condition of isometric contraction, and then stimulated at various frequencies for 20 min. The steady state values were obtained and normalized to the given stimulus frequency, as indicated in the graph (filled circles). *A)* developed force. Experimental data using human cardiac muscle were from Schwinger et al. (1993) (open circles) and from Pieske et al. (1999) (open triangles). *B)* diastolic force. Experimental data using human cardiac muscle were from Schmidt et al. (1998) (open circles). *C)* peak $Ca^{2+}_{cyt}$ concentration. Experimental data using human cardiac muscle were from Schmidt et al. (1998) (open circles) and from Pieske et al. (1995) (open triangles). *D)* diastolic $Ca^{2+}_{cyt}$ concentration. Experimental data using human cardiac muscle were from Schmidt et al. (1998) (open circles). *E)* $Na^{+}_{cyt}$ concentration. Experimental data using human cardiac muscle were from Pieske et al. (2002) (open circles).

namely preload increase, and (3) $\beta$-adrenergic receptor stimulation mimicking contractility increase, according to the literature on exercise in humans (Kang et al., 2017; Kurata et al., 2019; Lind & McNicol, 1967; Veldkamp et al., 2001; see also textbook by Herring & Paterson, 2018).

As a starting resting condition, the model was stimulated with a cycle length of 1 s, i.e. a heart rate of 60 min$^{-1}$, with isotonic contraction. Under the resting condition, diastolic hsmL was 0.950 $\mu$m without applying $F_{ext}$. The standard exercise condition was set as cycle length shortening to 0.33 s, i.e. heart rate increase to 180 min$^{-1}$, $F_{ext}$ increase to 5.00, i.e. diastolic hsmL elongation to 1.050 $\mu$m, and $\beta$-adrenergic receptor stimulation with the increased amplitude factors for L-type Ca$^{2+}$ current ($I_{CaL}$), SERCA, $I_{NaK}$, slowly activating K$^{+}$ current ($I_{Ks}$) as 1.30, 1.30, 1.20 and 12 times, respectively, plus the negative voltage shift of $I_{Ks}$ activation gate as $-5.0$ mV. This setting of $\beta$-adrenergic receptor stimulation was adapted from Kurata et al. (2019), with minor modifications especially for $I_{Ks}$. Since the contribution of $I_{Ks}$ to repolarization was set extremely low in the original Grandi model, it was necessary to increase the amplitude factor by 12 times and to shift the voltage by $-5.0$ mV, to obtain a comparable action potential configuration. The interventions were applied gradually with a time constant of 40 s, based on the literature on the heart rate change during exercise in humans (Lind & McNicol, 1967; Lemire et al., 2021).

In the simulations of Fig. 9, target cycle length, target $F_{ext}$ value, and extent of $\beta$-adrenergic receptor stimulation were varied to obtain various extent of exercise. The $F_{ext}$ of 0.00–10.00 correspond to diastolic hsmLs of 0.950–1.100 $\mu$m when stimulated with a cycle length of 1 s. Configurations of hsmL shortenings with various $F_{ext}$ values are presented in the Appendix Fig. A4. The extent of $\beta$-adrenergic receptor stimulation was changed using the following terms:

For $I_{CaL}$:

$$AI_{CaL} = 1 + \alpha \times 0.3,$$

$$P_{Ca} = AI_{CaL} \times P_{Ca,0,}$$

$$P_{Na} = AI_{CaL} \times P_{Na,0},$$

$$P_{K} = AI_{CaL} \times P_{K,0},$$

where $P_{Ca}$, $P_{Na}$ and $P_{K}$ are permeability factors for Ca$^{2+}$, Na$^{+}$ and K$^{+}$, respectively, and $P_{Ca,0}$, $P_{Na,0}$ and $P_{K,0}$ are permeability factors in the absence of $\beta$-adrenergic receptor stimulation for Ca$^{2+}$, Na$^{+}$ and K$^{+}$, respectively.

For SERCA:

$$ASERCA = 1 + \alpha \times 0.3,$$

$$V_{max\_SERCA} = ASERCA \times V_{max\_SERCA,0},$$

where $V_{max\_SERCA}$ is maximum velocity and $V_{max\_SERCA,0}$ is maximum velocity in the absence of $\beta$-adrenergic receptor stimulation.

For $I_{NaK}$:

$$AI_{NaK} = 1 + \alpha \times 0.2,$$

$$V_{max\_INaK} = AI_{NaK} \times V_{max\_INaK,0},$$

where $V_{max\_INaK}$ is maximum velocity and $V_{max\_INaK,0}$ is maximum velocity in the absence of $\beta$-adrenergic receptor stimulation.

For $I_{Ks}$:

$$AI_{Ks} = 1 + \alpha \times 11,$$

$$G_{Ks} = AI_{Ks} \times G_{Ks,\,0},$$

$$V\text{shift} = \alpha \times 5.0,$$

$$x\text{Ks}_{ss} = \frac{1}{1 + \exp\left(-\frac{Vm+3.8+V\text{shift}}{14.25}\right)},$$

$$\tau_{xKs} = \frac{990.1}{1 + \exp\left(-\frac{Vm+2.436+V\text{shift}}{14.12}\right)},$$

where $G_{Ks}$, $G_{Ks,0}$, $V$shift, $x$Ks$_{ss}$ and $\tau_{xKs}$ are conductance, conductance in the absence of $\beta$-adrenergic receptor stimulation, voltage shift factor in the presence of $\beta$-adrenergic receptor stimulation, steady-state value of gate and time constant of gate, respectively.

For the $\beta$-adrenergic receptor stimulation, $\alpha$ value is varied from 0 to 3.0. In the standard exercise condition, $\alpha = 1.0$.

In order to analyse the effects of Ca$^{2+}{}_{mit}$-dependent activations of three dehydrogenases, pyruvate dehydrogenase (PDHC), isocitrate dehydrogenase (ICDH) and oxoglutarate dehydrogenase (OGDH), simulations of 'silenced Ca$^{2+}{}_{mit}$-dependent regulation' were performed. In these simulations, the following allosteric regulation terms were set to refer to the average Ca$^{2+}{}_{mit}$ concentration per stimulation cycle prior to exercise, expressed as FCa$_{mit}$.

For PDHC:

$$f_{PDHa} = \left(1 + u_2 \times \left(1 + \frac{u_1 \times K_{Ca}{}^{nCa}}{K_{Ca}{}^{nCa} + FCa_{mit}{}^{nCa}}\right)\right)^{-1},$$

where $f_{PDHa}$, $K_{Ca}$ and $n$Ca are activation factor by Ca$^{2+}{}_{mit}$, binding constant of Ca$^{2+}{}_{mit}$, and Hill coefficient, respectively. $u_1$ and $u_2$ are model fitted factors.

For ICDH:

$$Ca_{Act1\_1} = 1 + \frac{\alpha_{ADP} \times \beta_{Ca1} \times FCa_{mit}}{\alpha_{Ca1} \times \beta_{ADP} \times K_{Ca1} \times \left(1 + \left(\frac{[Mg^{2+}]_{mit}}{K_{iMg2}}\right)^2\right)},$$

$$Ca_{Act1\_2} = 1 + \frac{\alpha_{ADP} \times FCa_{mit}}{\alpha_{Ca1} \times K_{Ca1} \times \left(1 + \left(\frac{[Mg^{2+}]_{mit}}{K_{iMg2}}\right)^2\right)},$$

$$Ca_{Act1\_3} = 1 + \frac{FCa_{mit}}{K_{Ca1} \times \left(1 + \left(\frac{[Mg^{2+}]_{mit}}{K_{iMg2}}\right)^2\right)},$$

$$Ca_{Act2\_1} = 1 + \frac{\alpha_{ATP} \times \beta_{Ca2} \times FCa_{mit}}{\alpha_{Ca2} \times \beta_{ATP} \times K_{Ca2} \times \left(1 + \left(\frac{[Mg^{2+}]_{mit}}{K_{iMg2}}\right)^2\right)},$$

$$Ca_{Act2\_2} = 1 + \frac{\alpha_{ATP} \times FCa_{mit}}{\alpha_{Ca2} \times K_{Ca2} \times \left(1 + \left(\frac{[Mg^{2+}]_{mit}}{K_{iMg2}}\right)^2\right)},$$

$$Ca_{Act2\_3} = 1 + \frac{FCa_{mit}}{K_{Ca2} \times \left(1 + \left(\frac{[Mg^{2+}]_{mit}}{K_{iMg2}}\right)^2\right)},$$

where $Ca_{ACT}$, $K_{Ca}$ and $K_{iMg2}$ are activation factors by $Ca^{2+}_{mit}$, binding constants of $Ca^{2+}_{mit}$, and binding constant for inhibition by $Mg^{2+}_{mit}$, respectively. $\alpha$ and $\beta$ are model fitted factors.

For OGDH:

$$Ca_{Act1} = 1 + \left(\frac{FCa_{mit}}{K_{Ca}}\right)^{nCa},$$

$$Ca_{Act2} = 1 + \left(\frac{FCa_{mit}}{\alpha_{Ca} \times K_{Ca}}\right)^{nCa},$$

$$Ca_{Act3} = 1 + \beta_{Ca} \times \left(\frac{FCa_{mit}}{\alpha_{Ca} \times K_{Ca}}\right)^{nCa},$$

where $Ca_{ACT}$, $K_{Ca}$ and $nCa$ are activation factors by $Ca^{2+}_{mit}$, binding constant of $Ca^{2+}_{mit}$, and Hill coefficient, respectively. $\alpha$ and $\beta$ are model fitted factors.

For the control simulations with allosteric $Ca^{2+}_{mit}$-dependent regulations, $Ca^{2+}_{mit}$ was used instead of $FCa_{mit}$ (Saito et al., 2016). Full equations are summarized in the Online Supplementary Material Table S9.

## Results

### Characteristics of the integrated human ventricular cell model in the isotonic contraction

To simulate cardiac workload transitions during exercise, the model cell was stimulated with various cycle lengths under the condition of isotonic contraction as shown later. The steady state action potential configurations, $Ca^{2+}_{cyt}$ transients, $Ca^{2+}_{mit}$, active force generations, and hsmL shortenings stimulated with cycle lengths of 1, 0.5 and 0.33 s are shown in the Fig. 4. Shortening the cycle length from 1 to 0.5 s increased amplitude of $Ca^{2+}_{cyt}$ transients, resulting in larger developed force and hsmL shortening, though the extents were small. Further cycle length shortening to 0.33 s augmented diastolic $Ca^{2+}_{cyt}$ and force levels, resulting in shortened diastolic hsmL. Accordingly, the developed force became smaller with cycle lengths of 0.33 s, which was more noticeable than that seen in the isometric contraction (see Fig. 3). The beat-to-beat oscillation of $Ca^{2+}_{mit}$ was small, i.e. the amplitude was 60.9 nM, when stimulated with cycle length of 1 s. The shorter cycle length induced accumulation of $Ca^{2+}_{mit}$, which was caused by fast and slow $Ca^{2+}$ fluxes of CaUni and $NCX_{mit}$, respectively. These characteristics are comparable to experimental data for rat and rabbit ventricular myocytes, as well as human atrial myocytes (Lu et al., 2013; Mason et al., 2020; Wust et al., 2017).

Energetics-related parameters are summarized in the Table 1. In the Integrated Human Ventricular Cell Model, ATP was consumed by contraction (62.99%), SERCA (28.59%), $I_{NaK}$ (7.75%) and $I_{pCa}$ (0.67%), when stimulated with cycle length of 1 s. These values are close to experimental estimates made in guinea-pig ventricular muscle (Schramm et al., 1994): contraction (76%), SERCA (15%) and $I_{NaK}$ (9%). The phosphocreatine (PCr)/ATP concentration ratio was 2.33, which is comparable to experimental values 1.9–2.1 in the human hearts measured by ³¹P cardiac magnetic resonance spectroscopy (Phan et al., 2009; Smith et al., 2006).

The cycle length shortenings from 1 to 0.5 and 0.33 s increased $mVO_2$ by 1.8 and 2.6 times, respectively. While cytoplasmic ATP ($ATP_{cyt}$) concentration was hardly affected, cytoplasmic ADP ($ADP_{cyt}$) and Pi ($Pi_{cyt}$) concentrations increased and cytoplasmic PCr ($PCr_{cyt}$) concentration decreased. Steady state $NADH_{mit}$ level was maintained within a 10% change, comparable to an experimental report using rabbit hearts (Heineman & Balaban, 1993).

To study contribution of individual component to $mVO_2$ and $NADH_{mit}$, sensitivity analyses were performed. As expected from the linear relationship between $mVO_2$ and workload, the $Ca^{2+}_{cyt}$ handling components had great influence on $mVO_2$ (Fig. 5A). CaUni_js had a relatively greater influence on $mVO_2$, because it highly affected $Ca^{2+}_{mit}$ and $Ca^{2+}_{SR}$ concentrations, as will be demonstrated later. Regarding $NADH_{mit}$, the NADH-producing enzymes ICDH and OGDH had some influence on $NADH_{mit}$ levels (Fig. 5B). The factors involved in mitochondrial $H^+$ dynamics, such as electron transport chain, ATP synthase, phosphate carrier (PiC), $K^+$-$H^+$-exchanger (KHE) and citrate synthase (CS) were influential. Complex I was less influential, possibly because it is not the rate-limiting factor of electron transport chain (Dalmonte et al., 2009).

### Validation of stimulus frequency dependences of $NADH_{mit}$, $Ca^{2+}_{mit}$, force and hsmL

To further validate the Integrated Human Ventricular Cell Model, we next focused on stimulus

frequency-dependent, dynamic change of NADH$_{mit}$, which was experimentally observed in rat ventricular trabeculae (Brandes & Bers, 1997, 2002) and in guinea-pig ventricular myocytes (Jo et al., 2006). The model was first stimulated at 0.25 Hz under isometric (Fig. 6*A*) or isotonic (Fig. 6*B*) conditions, then stimulus frequency was abruptly increased to 1 or 2 Hz for 200 s. The model well reproduced the experimentally observed biphasic NADH$_{mit}$ changes, i.e. decrease followed by increase, and Ca$^{2+}$$_{mit}$ accumulation (Brandes & Bers, 1997, 2002; Jo et al., 2006). The NADH$_{mit}$ levels are determined by the balance between consumption via Complex I of the electron transport chain and production via four dehydrogenases, PDHC, ICDH, OGDH and malate dehydrogenase (MDH). As the stimulus frequency increased, both NADH consumption and production fluxes increased (left panels of Fig. 6*C, D*), with the consumption fluxes increasing at earlier timing as evidenced by the flux difference plots (right panels of Fig. 6*C, D*), causing the biphasic NADH$_{mit}$ changes

(Fig. 6*A, B*). The model also demonstrated greater shortenings of hsmL upon stimulus frequency increase with isotonic contraction (Fig. 6*B*), also comparable to experimental data (Jo et al., 2006). The generation of larger developed forces at higher stimulus frequencies (Fig. 6*A, B*) is a characteristic of human ventricular trabeculae (see also Fig. 3) that is distinct from rat ventricular trabeculae (Brandes & Bers, 1997, 2002; Maier et al., 2000).

Taken together, we conclude that the Integrated Human Ventricular Cell Model well reproduces the characteristics of excitation-contraction-energetics coupling of human ventricular myocytes.

## Simulation of exercise-induced responses of cardiac energetics

Changes in cardiac energetics during dynamic workload transitions and the underlying mechanisms were explored

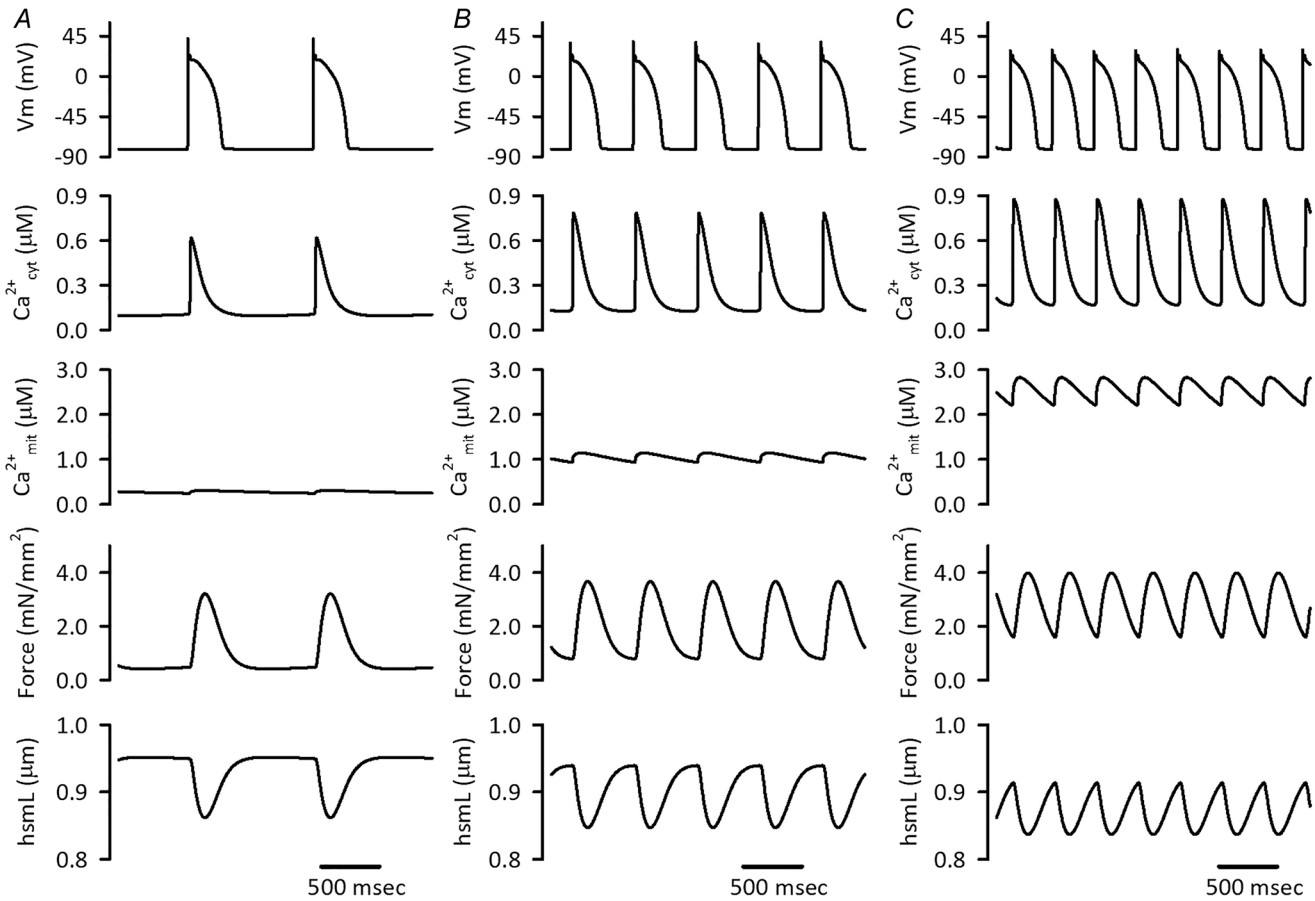

**Figure 4. Configurations of action potential, Ca$^{2+}$$_{cyt}$, Ca$^{2+}$$_{mit}$, force and hsmL shortenings stimulated with cycle lengths of 1–0.33 s in the Integrated Human Ventricular Cell Model**
The steady state variables stimulated with cycle lengths of 1 s (*A*), 0.5 s (*B*) and 0.33 s (*C*) were plotted. The initial condition was obtained without applying $F_{ext}$, i.e. 0.950 $\mu$m diastolic hsmL, under the condition of isotonic contraction stimulated with cycle length of 1 s for 30 min, as shown in the Online Supplementary Material Table S14.

using this model. During exercise, cardiac workload dramatically increases due to increased heart rate, preload and contractility. The *in vivo* exercise condition was established based on the literature, as described in the Methods section.

Upon exercise application, $NADH_{mit}$ initially decreased to 1.01 mM and then increased to 1.19 mM, similar to the level before the exercise onset. $Ca^{2+}_{mit}$, $Ca^{2+}_{cyt}$ and forces at diastole, systole, as well as developed, all increased during exercise (Fig. 7*A*). Among them, the $Ca^{2+}_{mit}$ increase was drastic; i.e. the diastolic $Ca^{2+}_{mit}$ increased by 13.5-fold. The diastolic hsmL initially

elongated because of the increase in preload, i.e. $F_{ext}$, but gradually shortened due to the subsequent increases in diastolic $Ca^{2+}_{cyt}$. Throughout exercise, the magnitude of cell shortening increased. Following the increases in ATP-consuming processes, $mVO_2$ increased by 3.35 times during exercise (Fig. 7*E*). $ATP_{cyt}$ was maintained nearly constant, at the expense of a decrease in $PCr_{cyt}$ (Fig. 7*F, G*). When $Ca^{2+}_{mit}$-dependent regulations of PDHC, ICDH and OGDH were silenced, $NADH_{mit}$ change shifted downward (grey dashed line in Fig. 7*A*), suggesting that an exercise-induced $Ca^{2+}_{mit}$ increase contributes to maintaining $NADH_{mit}$. On the other hand, silencing

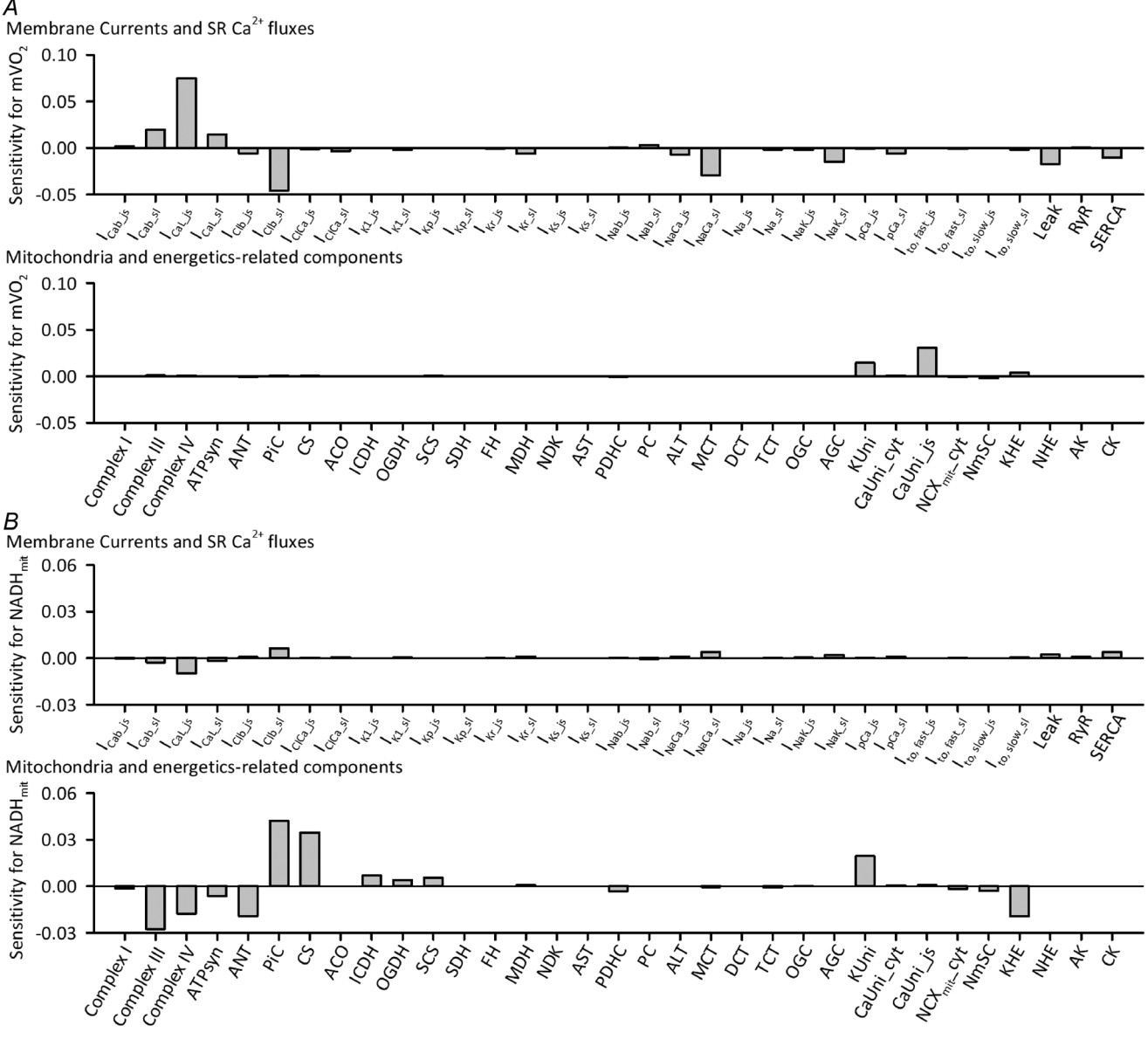

**Figure 5. Sensitivity analyses of the Integrated Human Ventricular Cell Model**
The model was stimulated with a cycle length of 1 s with ± 5% change of the amplitude factor for each component for 20 min. The initial condition was the same as in Fig. 4. Sensitivity was calculated as the relative change of $mVO_2$ (*A*) and $NADH_{mit}$ (*B*) as follows: Sensitivity $= \frac{(X_{+5\%} - X_{-5\%})}{X_{original}}$.

the regulation did not alter mVO$_2$, ATP$_{cyt}$ and PCr$_{cyt}$; note that the time courses with silenced regulation (grey lines in Fig. 7*A, E–G*) almost overlapped with those of controls. Upon exercise application, both NADH$_{mit}$ consumption and production fluxes increased (Fig. 8*A, B*), with the consumption flux increasing earlier, as revealed by the flux difference plot (Fig. 8*C*). This caused biphasic NADH$_{mit}$ change (Fig. 7*A*), as seen with the stimulus frequency increase protocols (see Fig. 6). Silencing the Ca$^{2+}_{mit}$-dependent regulation of dehydrogenases shifted the flux difference plot downward during the early phase of the exercise (grey dashed line in Fig. 8*C*), making the NADH$_{mit}$ change downward (grey dashed line in Fig. 7*A*).

The contribution of three exercise factors, namely heart rate, preload and contractility, to cardiac energetics was dissected by applying one factor at a time: cycle

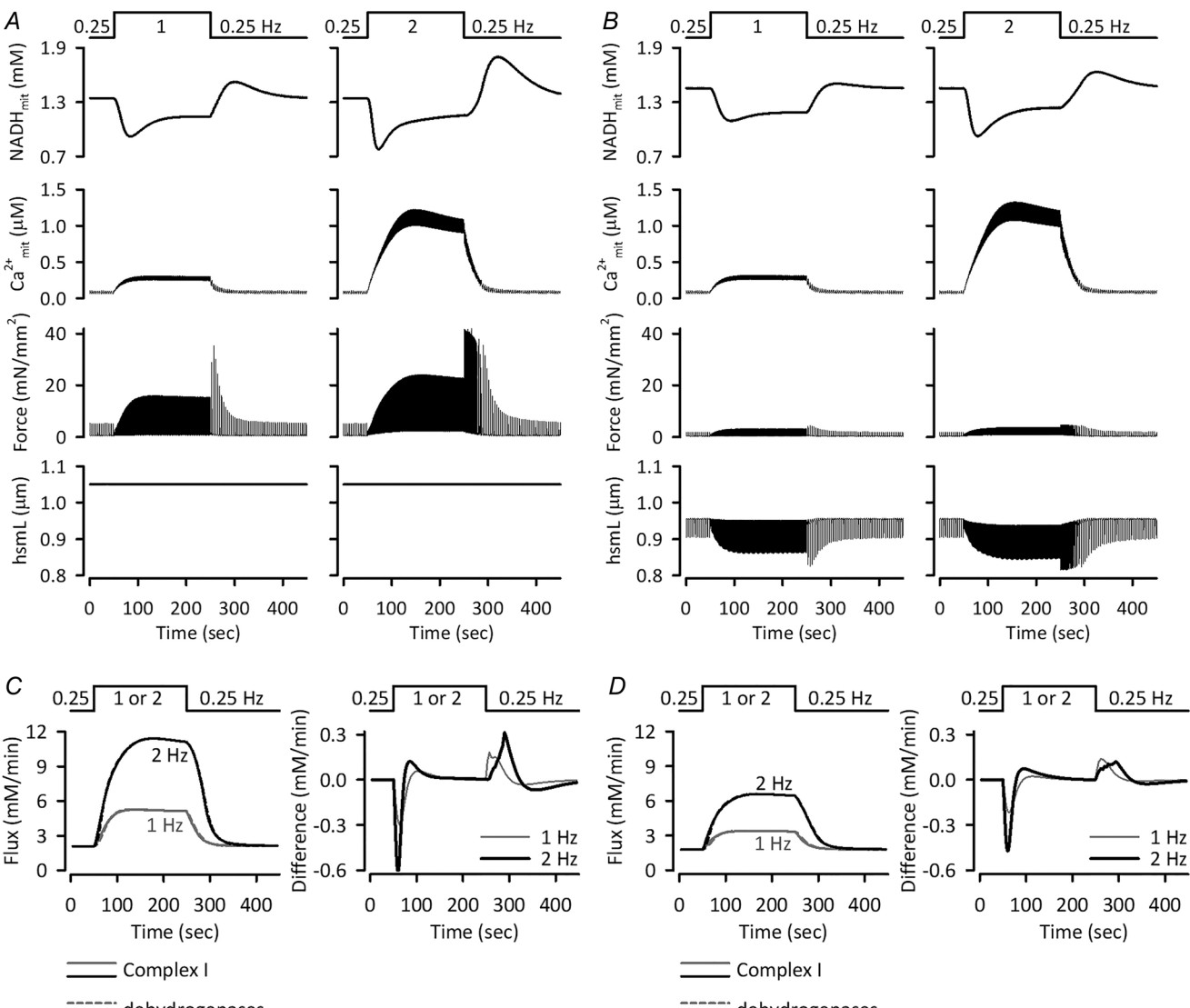

**Figure 6. Responses of NADH$_{mit}$, Ca$^{2+}_{mit}$, force and hsmL in the Integrated Human Ventricular Cell Model to an abrupt increase of the stimulus frequency**

*A*) the Integrated Human Ventricular Cell Model was applied with $F_{ext}$ of 4.25, i.e. 1.050 $\mu$m diastolic hsmL, under the condition of isometric contraction, and was then stimulated with cycle length of 4 s, i.e. at 0.25 Hz, for 20 min to obtain steady state. Abrupt increases in the stimulus frequency to 1 Hz (cycle length 1 s, left panels) and 2 Hz (cycle length 0.5 s, right panels) were applied at 50 s for 200 s. *B*) the model with the initial condition as in Fig. 4, isotonic contraction with $F_{ext}$ of 0.00, was stimulated at 0.25 Hz for 20 min to obtain steady state. Simulation protocol was the same as in *A*. *C* and *D*) NADH-consuming flux of Complex I (continuous lines) and NADH-producing fluxes of all dehydrogenases, PDHC, ICDH, OGDH and MDH (dashed lines) (left panels), and the difference, NADH-producing flux minus NADH-consuming fluxes (right panels) in the simulations shown in *A* and *B*, respectively.

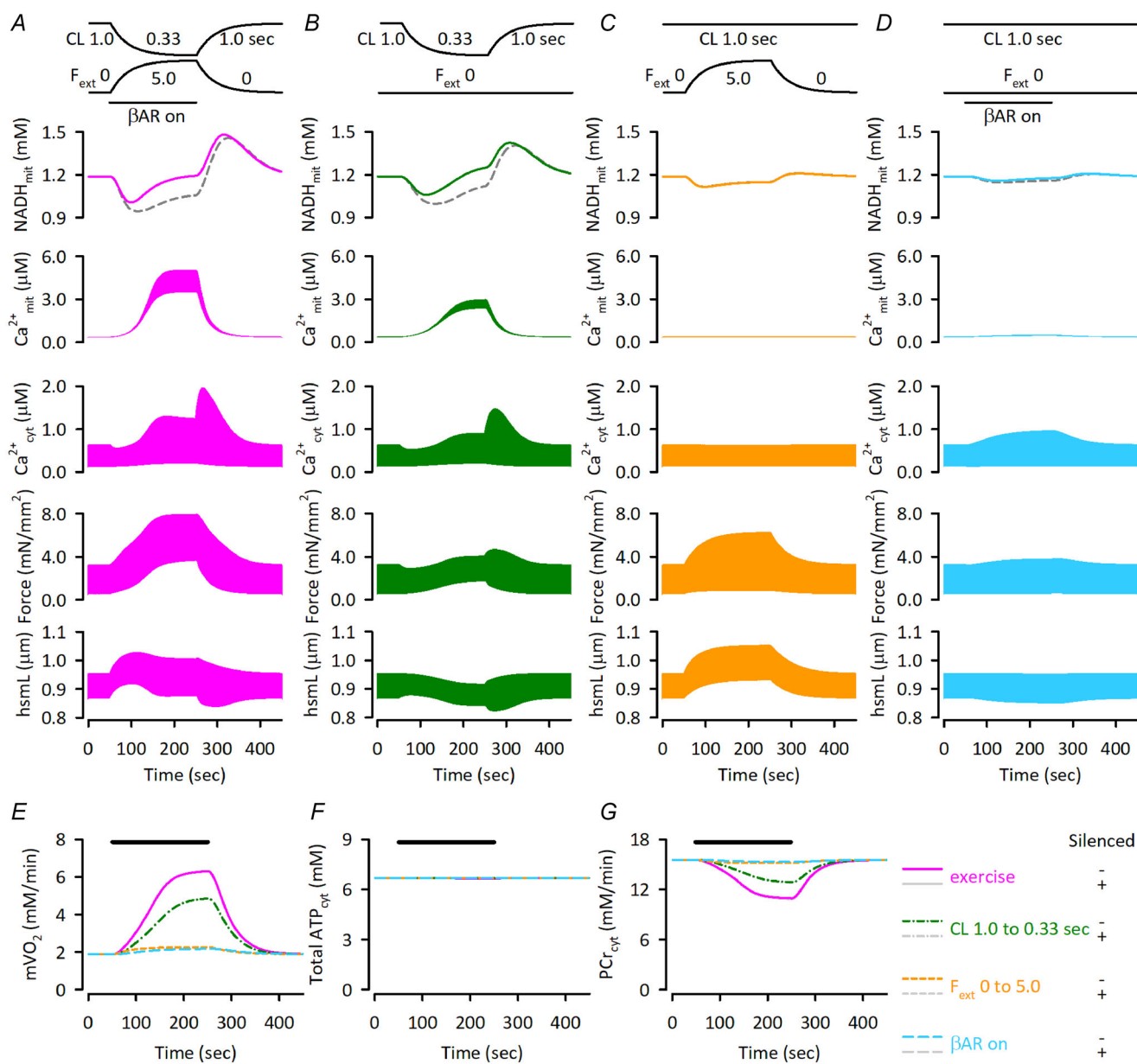

**Figure 7. Responses of the Integrated Human Ventricular Cell Model to dynamic changes of workload by various interventions**

The dynamic changes of workload were applied gradually at 50 s for 200 s with a time constant of 40 s. *A)* responses of NADH$_{mit}$, Ca$^{2+}$$_{mit}$, force and hsmL without (control; magenta continuous lines) or with (dashed grey lines) silenced Ca$^{2+}$$_{mit}$-dependent regulation to exercise protocol, composed of cycle length (CL) shortening to 0.33 s, $F_{ext}$ increase to 5.0, and application of $\alpha = 1.0$ for $\beta$-adrenergic stimulation. *B)* the responses when only cycle length was shortened to 0.33 s. *C)* the responses when only $F_{ext}$ was increased to 5.0. *D)* the responses when only $\beta$-adrenergic stimulation was applied, $\alpha = 1.0$. The dashed grey lines are with silenced Ca$^{2+}$$_{mit}$-dependent regulation (*B–D*). *E–G*) responses of mVO$_2$ (*E*), total ATP$_{cyt}$ concentration (*F*), and PCr$_{cyt}$ concentration (*G*) to dynamic workload changes during exercise (magenta continuous lines), cycle length shortening to 0.33 s (green dotted-dashed lines), $F_{ext}$ increase to 5.0 (orange dashed lines), and application of $\alpha = 1.0$ for $\beta$-adrenergic stimulation (sky-blue long dashed lines). Simulation results with silenced Ca$^{2+}$$_{mit}$-dependent regulation are shown as grey lines, though they overlap with those with the control regulations. The initial condition of the model was the same as in Fig. 4.

length shortening (Fig. 7*B*), $F_{ext}$ increase (Fig. 7*C*), and $\beta$-adrenergic receptor stimulation (Fig. 7*D*). Among the three factors, cycle length shortening had the greatest effect on $NADH_{mit}$, $Ca^{2+}_{mit}$, $mVO_2$ and $PCr_{cyt}$. Since the increase in ATP consumption in a given period, i.e. 1 min, was much greater with cycle length shortening (Fig. 7*B*) than other factors, $mVO_2$ was most affected by cycle length shortening (Fig. 7*E*), consistent with the well-accepted idea that heart rate is the major determinant of $mVO_2$ during exercise (see textbook by Herring & Paterson, 2018). Along with the $mVO_2$ increase, flux of Complex I increased to consume $NADH_{mit}$ (see Fig. 8*A*); thus the initial $NADH_{mit}$ drop was most affected by cycle length shortening. Cycle length shortening and $\beta$-adrenergic receptor stimulation increased $Ca^{2+}_{cyt}$ to a similar extent, but the $Ca^{2+}_{mit}$ increase, 9.03-fold, was much greater with cycle length shortening than with $\beta$-adrenergic receptor stimulation, 1.38-fold. An increase in $F_{ext}$ had little effect on $Ca^{2+}_{cyt}$, so $Ca^{2+}_{mit}$ remained unchanged. Accordingly, the contribution of $Ca^{2+}_{mit}$-dependent regulation to increasing $NADH_{mit}$ from the initial drop mostly depended on cycle length shortening.

Next, the three factors were varied in different combinations to simulate various extent of exercise, and the parameters 200 s after the application of the factors were obtained (Fig. 9). The cycle length shortenings from 1.00 to 1.00–0.33 s with (green filled circles) and without (green open circles) $F_{ext}$ increase to 5.0 plus standard $\beta$-adrenergic receptor stimulation ($\alpha = 1.0$) covered a wide 3.35-fold $mVO_2$ change (Fig. 9*A*). Changing $F_{ext}$ (orange triangles) and $\beta$-adrenergic receptor stimulation (sky-blue squares) did not produce as large an $mVO_2$ difference as changing cycle length, regardless of whether

it was combined with other factors (compare filled *vs.* open symbols). Meanwhile, the $NADH_{mit}$ concentration at the end of various exercise regimes was 1.14–1.25 mM, −4.2 to +5.3% ($n = 38$, standard deviation (SD) = 2.6%) of that before starting exercise. For this small $NADH_{mit}$ change, the increase in $F_{ext}$ had a relatively large effect; see the steeper change of $NADH_{mit}$ per $mVO_2$ (orange triangles) than the one for different cycle lengths (green circles) or different $\alpha$ values in $\beta$-adrenergic receptor stimulation (sky-blue squares). Interestingly, in the $mVO_2$ range of 2.5–5.0 mM/min, $NADH_{mit}$ became larger with larger $mVO_2$. This is due to the $Ca^{2+}_{mit}$-dependent regulation of dehydrogenases (Fig. 9*B*). The $NADH_{mit}$ concentration with different $F_{ext}$ values aligned along the vertical lines, since stretch did not affect $Ca^{2+}_{cyt}$ nor $Ca^{2+}_{mit}$ but increased ATP consumption by contraction (see orange triangles in Fig. 9*B*). On the other hand, with different cycle lengths and different $\beta$-adrenergic receptor stimulations, $NADH_{mit}$ concentration showed three phases as $Ca^{2+}_{mit}$ increased: decrease, increase and decrease (see green circles and sky-blue squares in Fig. 9*B*). At $Ca^{2+}_{mit} < \sim 400$ nM, $NADH_{mit}$ decreased as $Ca^{2+}_{mit}$ increased because of increased ATP consumption by SERCA, $I_{pCa}$ and contraction induced by larger $Ca^{2+}_{cyt}$. At $Ca^{2+}_{mit}$ of $\sim 400$ nM–$\sim 2.2$ μM, $NADH_{mit}$ increased because of $Ca^{2+}_{mit}$-dependent activation of dehydrogenases; note that the $Ca^{2+}$ affinity for PDHC, ICDH and OGDH are around several hundred nanomolar (see Online Supplementary Material Table S9 and McCormack et al., 1990; Rutter & Denton, 1988). The third $NADH_{mit}$ decrease phase at $Ca^{2+}_{mit} > \sim 2.2$ μM was again a result of increased ATP consumption by SERCA, $I_{pCa}$ and contraction, because dehydrogenase activities were saturated. Since an increase in $mVO_2$ reflects an

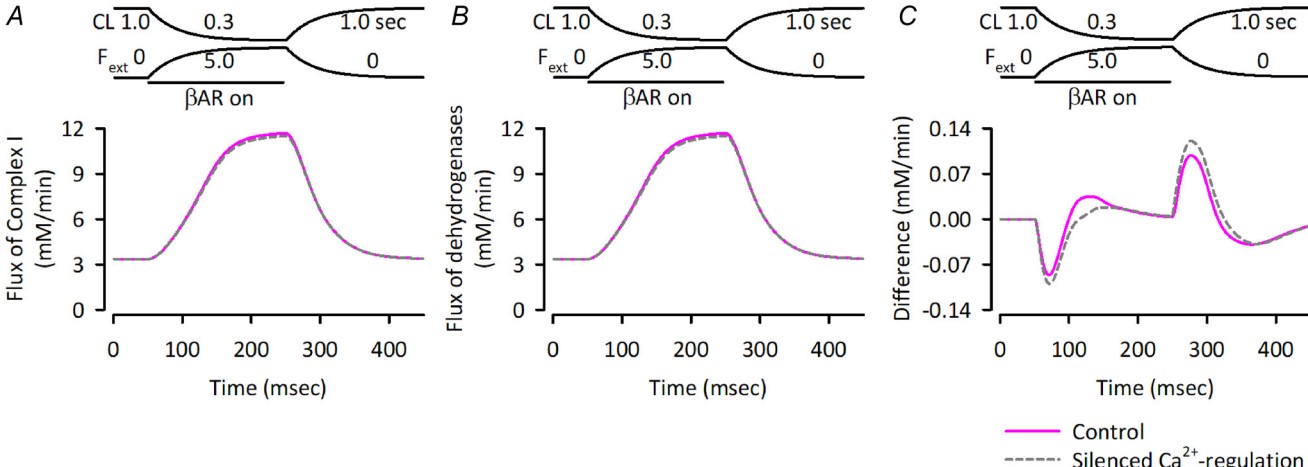

**Figure 8. Responses of the NADH-consuming and producing fluxes in the Integrated Human Ventricular Cell Model to exercise**
Responses of NADH-consuming flux of Complex I (*A*), NADH-producing fluxes of all dehydrogenases, PDHC, ICDH, OGDH and MDH (*B*), and the difference (*C*) without (control; magenta continuous lines) or with (dashed grey lines) silenced $Ca^{2+}_{mit}$-dependent regulation. The initial condition and the exercise protocol were the same as in Fig. 7*A*.

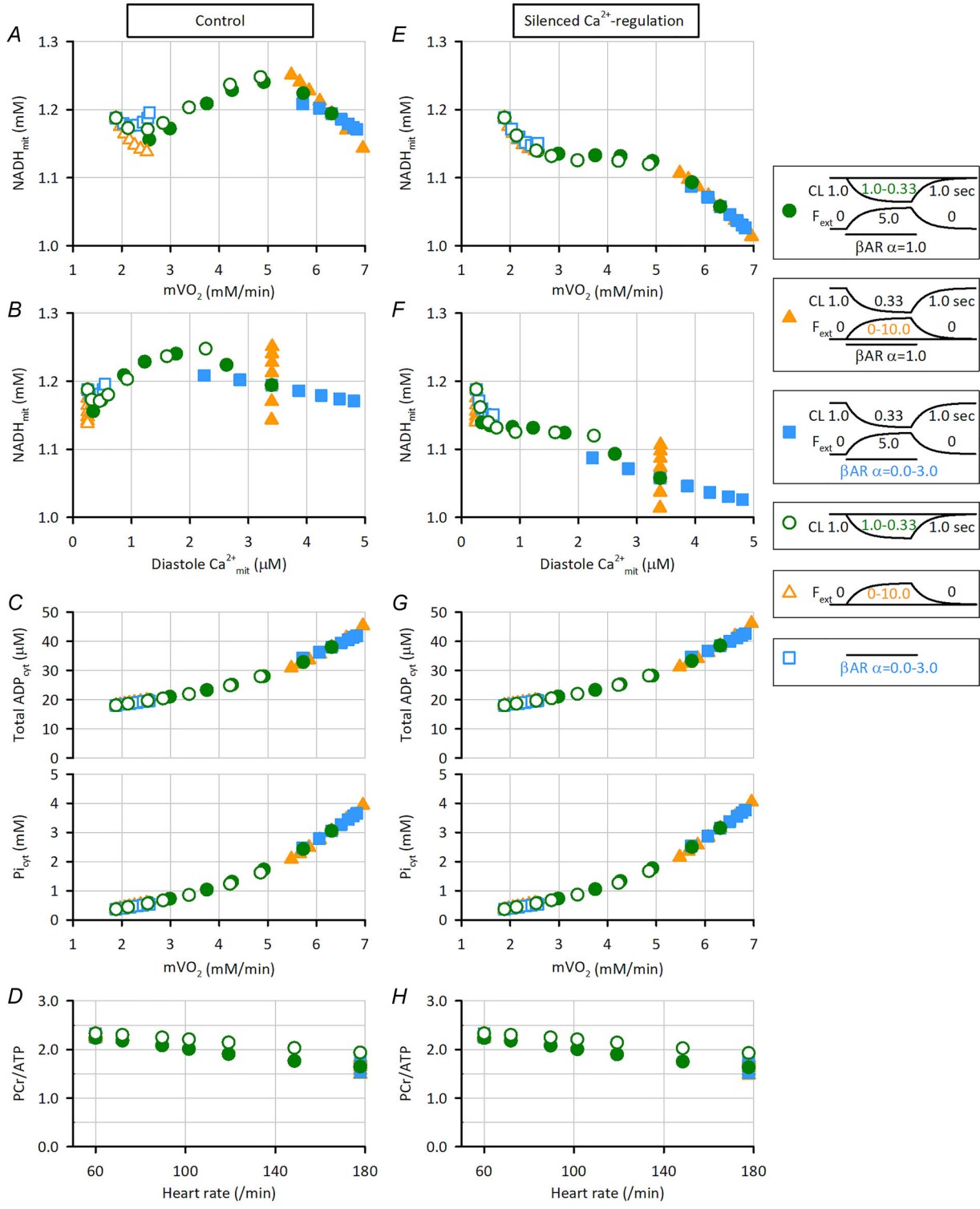

**Figure 9. Dependence of NADH$_{mit}$, total ATP$_{cyt}$ and Pi$_{cyt}$ on various workloads**

The workload increase by various factors was applied at 50 s with a time constant of 40 s. Values at 200 s after the onset of the workload increase were obtained without (*A–D*) or with (*E–H*) silenced Ca$^{2+}_{mit}$-dependent regulation. Interventions were: cycle length (CL) shortening from 1.00 to 1.00, 0.83, 0.67, 0.59, 0.50, 0.40, 0.33 s with (green filled circles) or without (green open circles) $F_{ext}$ increase to 5.00 plus application of $\alpha = 1.0$ for $\beta$-adrenergic stimulation, $F_{ext}$ increase from 0.00 to 0.00, 0.90, 1.98, 3.34, 5.00, 7.25 and 10.00 with (orange filled triangles) or without (orange open triangles) cycle length shortening to 0.33 s plus application of $\alpha = 1.0$ for $\beta$-adrenergic stimulation, and application of $\alpha = 0.0, 0.5, 1.0, 1.5, 2.0, 2.5, 3.0$ for $\beta$-adrenergic stimulation with (sky-blue filled

squares) or without (sky-blue open squares) cycle length shortening to 0.33 s plus $F_{ext}$ increase to 5.0. *A* and *E*) dependence of $NADH_{mit}$ on $mVO_2$. *B* and *F*) dependence of $NADH_{mit}$ on diastolic $Ca^{2+}_{mit}$. *C* and *G*) dependence of total $ATP_{cyt}$ (upper) or $Pi_{cyt}$ (lower) on $mVO_2$. *D* and *H*) dependence of PCr/ATP on heart rate. The initial condition was the same as in Fig. 4.

increase in ATP consumption, the products $ADP_{cyt}$ and $Pi_{cyt}$ increased as $mVO_2$ increased, as expected (Fig. 9*C*).

When the $Ca^{2+}_{mit}$-dependent regulation was silenced, $NADH_{mit}$ shifted to a lower level, and the $NADH_{mit}$ increasing phase disappeared (Fig. 9*E*, *F*). The overall $NADH_{mit}$ difference increased to 0.0 to −14.7% ($n = 38$, SD = 4.0%; concentration at the end of various exercise regimes was 1.19–1.01 mM) of that before starting exercise, which was still small but larger than the values of −4.2 to +5.3% in the control model. On the other hand, silencing the $Ca^{2+}_{mit}$-dependent regulation did not alter $ADP_{cyt}$, $Pi_{cyt}$ and PCr/ATP (Fig. 9*C vs. G*, *D*, *vs. H*).

These analyses revealed that the $Ca^{2+}_{mit}$-dependent regulation made a small but remarkable contribution to the maintenance of $NADH_{mit}$ during dynamic workload increases.

### Roles of uneven distributions of $Ca^{2+}_{mit}$ and $Ca^{2+}_{SR}$ handling proteins to cardiac energetics

The contribution of spatial and functional couplings of mitochondria and SR via CaUni-RyR and $NCX_{mit}$-SERCA were analysed using a non-MSI model. In this model, NmSC and CaUni_js were removed and all $NCX_{mit}$, CaUni and SERCA were set to face the cytoplasm (see Fig. 10). The basic characteristics of the non-MSI model with cycle length of 1 s are presented in the Table 1 and Appendix Fig. A5. Action potential configuration and most ionic currents were

not greatly affected (compare Appendix Fig. A2*A* and A5*A*). However, $Ca^{2+}_{cyt}$ transients, $Ca^{2+}_{mit}$, and therefore the extents of active force generation and hsmL shortening, as well as $Ca^{2+}$ flux via RyR were diminished (see Appendix Fig. A5Da, De, Eb, Ea and Ba), resulting in lower ATP consumption (Table 1). Accordingly, $mVO_2$ became smaller, $NADH_{mit}$ and $PCr_{cyt}$ became larger in the non-MSI model than in the MSI model (Table 1).

Then responses of cardiac energetics to dynamic workload transition were investigated. Exercise protocol application to the non-MSI model induced biphasic $NADH_{mit}$ change, an initial decrease followed by an increase (Fig. 11*A*), as observed in the control MSI model (see Fig. 7A). However, the decrease was larger and the increase was slower than in the MSI model, thus $NADH_{mit}$ did not return to its pre-exercise level despite the 1.88 times lower $mVO_2$ during exercise (Fig. 11*E*; compare with Fig. 7*E*). The lower $mVO_2$ in the non-MSI model was due to the smaller increases in $Ca^{2+}_{cyt}$ transients, $Ca^{2+}_{mit}$, active force generation and hsmL shortening (see Fig. 11*A*; compare with Fig. 7*A*). The effects of changing the three factors one at a time on $Ca^{2+}_{cyt}$ transients, active force generation and hsmL shortening were smaller than in the MSI model (Fig. 11*B-D*). Note that pre-exercise diastolic $Ca^{2+}_{mit}$ concentration in the non-MSI model was 36 nM, 6.91 times lower than that in the MSI model, and $Ca^{2+}_{mit}$ increased to only 57 nM during exercise, which was far below the $Ca^{2+}_{mit}$ affinity of dehydrogenases. Accordingly, silencing $Ca^{2+}_{mit}$-dependent regulation had a negligible effect in the non-MSI model.

Finally, the roles of the uneven distributions of $Ca^{2+}_{mit}$ and $Ca^{2+}_{SR}$ handling proteins on cardiomyocyte functions were investigated. The fractional ratios of CaUni_js and NmSC to total CaUni and $NCX_{mit}$, respectively, were systematically varied from 0.0–1.0 in 0.1 increments. The exercise protocols were then applied to the models after stimulation with a cycle length of 1 s for 20 min. Note that the combination of both CaUni_js and NmSC fractions as 0.0 corresponds to the non-MSI model described above (Figs 10, 11). It was clearly demonstrated that diastolic $Ca^{2+}_{mit}$ and $Ca^{2+}_{SR}$ concentrations strongly depend on CaUni_js fraction, regardless of exercise application (Fig. 12*A–D*). When the CaUni_js fractions were 0.0 and 0.1, i.e. the percentages of CaUni_js to total CaUni were 0% and 10%, respectively, the diastolic $Ca^{2+}_{mit}$ during exercise was below 400 nM, which was the threshold for the $Ca^{2+}_{mit}$-dependent regulation to become predominant, as shown in the Fig. 9*A* and *B*. The NmSC fraction also had positive relationships with $Ca^{2+}_{mit}$ and $Ca^{2+}_{SR}$ concentrations, though the impacts were

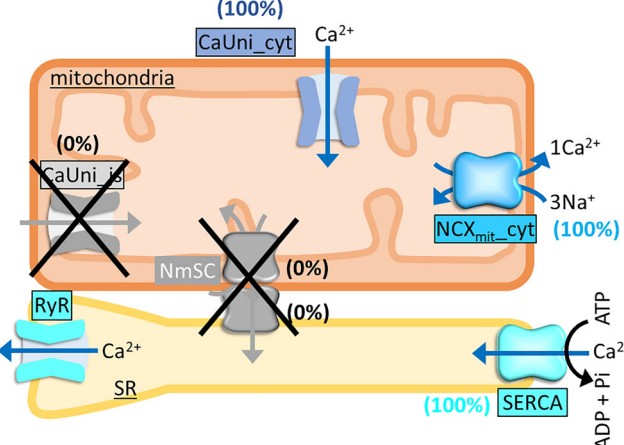

**Figure 10. Localization settings of mitochondrial and SR $Ca^{2+}$ handling proteins in the non-MSI model**
The fractions of CaUni_js and $NCX_{mit}$-SERCA complex NmSC were set as 0%, and those of CaUni_cyt, $NCX_{mit}$_cyt and SERCA as 100%.

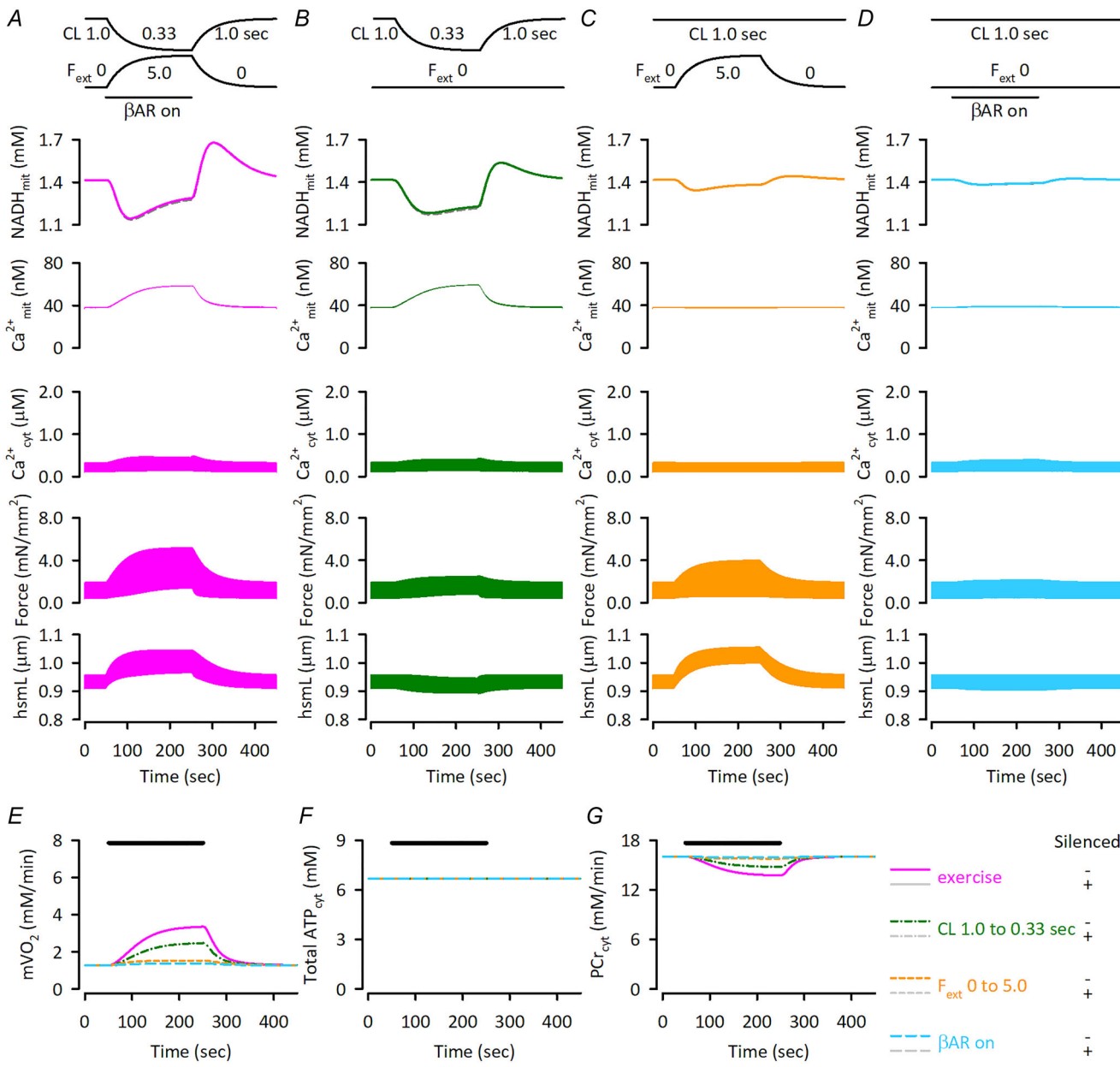

**Figure 11. Responses of the non-MSI model to dynamic changes of workload by various interventions**
*A*) responses of $NADH_{mit}$, $Ca^{2+}_{mit}$, force and hsmL without (control; magenta continuous lines) or with (dashed grey lines) silenced $Ca^{2+}_{mit}$-dependent regulation to exercise protocol, composed of cycle length (CL) shortening to 0.33 s, $F_{ext}$ increase to 5.0, and application of $\alpha = 1.0$ for $\beta$-adrenergic stimulation. *B*) the responses when only cycle length was shortened to 0.33 s. *C*) the responses when only $F_{ext}$ was increased to 5.0. *D*) the responses when only $\beta$-adrenergic stimulation was applied, $\alpha = 1.0$. The dashed grey lines are with silenced $Ca^{2+}_{mit}$-dependent regulation (*B*–*D*). *E*–*G*) responses of $mVO_2$ (*E*), total $ATP_{cyt}$ concentration (*F*), and $PCr_{cyt}$ concentration (*G*) to dynamic workload changes by exercise (magenta continuous lines), cycle length shortening to 0.33 s (green dotted-dashed lines), $F_{ext}$ increase to 5.0 (orange dashed lines), and application of $\alpha = 1.0$ for $\beta$-adrenergic stimulation (sky-blue long dashed lines). Simulation results with silenced $Ca^{2+}_{mit}$-dependent regulation are shown as grey lines, though they overlap with those with the control regulations. The initial condition of the non-MSI model, isotonic contraction without applying $F_{ext}$, is shown in the Online Supplementary Material Table S14. Then the same protocols were applied as in Fig. 7.

smaller than with the CaUni_js fraction (Fig. 12*A–D*). In order to evaluate the contribution of the uneven distributions of $Ca^{2+}_{mit}$ and $Ca^{2+}_{SR}$ handling proteins to the $Ca^{2+}_{mit}$-dependent regulation of dehydrogenases during exercise, the percentage of $NADH_{mit}$ recovery at the end of the exercise protocol from the initial drop was obtained (Fig. 13). For most combinations of CaUni_js and NmSC fractions, the $NADH_{mit}$ recovery was greater in the model with $Ca^{2+}_{mit}$-dependent regulation than in the one with the silenced regulation (Fig. 13*A*). The contribution of $Ca^{2+}_{mit}$-dependent regulation was expressed as the difference in $NADH_{mit}$ recovery between without and with silenced regulation. As expected, the contribution was minor for CaUni_js fractions of 0.0 and 0.1, since the $Ca^{2+}_{mit}$ concentration was below the threshold level for the regulation. The contribution also became smaller for larger fractions of CaUni_js. This is because $Ca^{2+}_{mit}$-dependent regulation of dehydrogenases was already saturated yet ATP-consuming processes were still increased by the increase in $Ca^{2+}_{cyt}$. Accordingly, the NADH production increase was obscured by the NADH

consumption increase. The maximum contribution of ~40% was obtained with a CaUni_js fraction of 0.4. The standard setting of the control MSI model – CaUni_js and NmSC fractions of 0.5 and 0.7, respectively, values that were adopted from experiments – showed near-maximal contribution, 36.9% (Fig. 13*B*).

## Discussion

It has been suggested that distinct spatial distributions of the $Ca^{2+}_{mit}$ handling proteins MCU and NCLX, and their coupling with the $Ca^{2+}_{SR}$ handling proteins RyR and SERCA, respectively, contribute to efficient SR-mitochondria $Ca^{2+}$ signalling in cardiomyocytes (De La Fuente et al., 2016, 2018; Takeuchi & Matsuoka, 2022). However, there was little information on how efficient the signalling is, nor on how it contributes to ventricular myocyte functions. In the present study, we developed a new Integrated Human Ventricular Cell Model considering the spatial and functional couplings of $Ca^{2+}_{mit}$ and $Ca^{2+}_{SR}$ handling proteins, and successfully

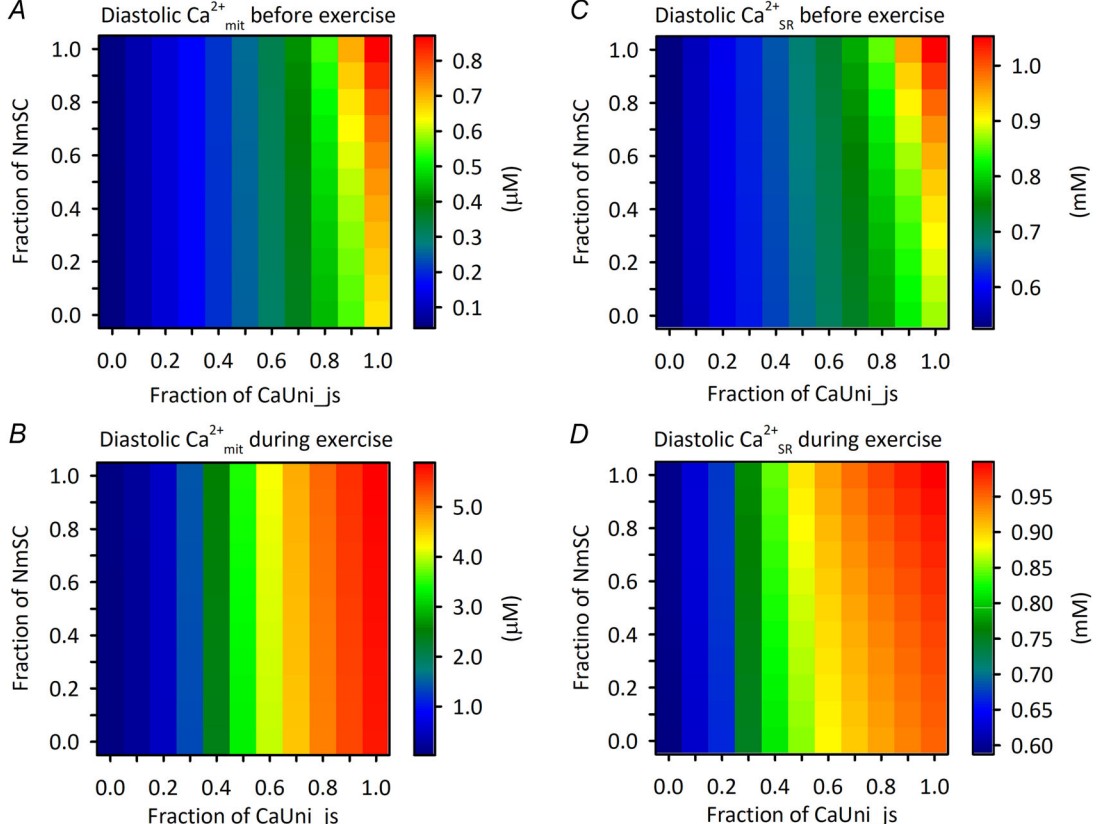

**Figure 12. Effects of varying fractional ratio of CaUni_js and NmSC on $Ca^{2+}_{mit}$ and $Ca^{2+}_{SR}$**
The fractional ratios of CaUni_js and NmSC were systematically varied from 0.0–1.0 in 0.1 increments. The model cell was stimulated with a cycle length of 1 s without applying $F_{ext}$ under the condition of isotonic contraction for 20 min, and then the same exercise protocol as in Figs 7*A* and 11*A* was applied. The values just before and 200 s after the exercise onset were obtained. *A*) diastolic $Ca^{2+}_{mit}$ just before exercise onset. *B*) diastolic $Ca^{2+}_{mit}$ 200 s after exercise onset. *C*) diastolic $Ca^{2+}_{SR}$ just before exercise onset. *D*) diastolic $Ca^{2+}_{SR}$ 200 s after exercise onset.

simulated excitation-contraction-energetics coupling. Quantitative model analyses revealed that uneven distributions of the handling proteins contribute to $Ca^{2+}_{mit}$ accumulation particularly during exercise, which promotes more stability of $NADH_{mit}$ through activating NADH-producing dehydrogenases. Furthermore, the analyses revealed that the uneven distribution of the handling proteins, uncovered by *in vitro* experiments (De La Fuente et al., 2016; Takeuchi & Matsuoka, 2022), optimizes the effect of $Ca^{2+}_{mit}$-dependent regulation of dehydrogenases to stabilize $NADH_{mit}$.

Among the three factors comprising exercise, changing cycle length drastically affected $Ca^{2+}_{mit}$, which was attributable to faster CaUni flux than $NCX_{mit}$ flux. That is, with longer cycle length, there was enough time left for the slower $NCX_{mit}$ to extrude more $Ca^{2+}_{mit}$. As the cycle length was shortened, the next stimulus came before sufficient amounts of $Ca^{2+}_{mit}$ was extruded, resulting in a staircase-like $Ca^{2+}_{mit}$ accumulation. Although it is still controversial whether $Ca^{2+}_{mit}$ oscillates beat-by-beat, most reports in the literature agree that $Ca^{2+}_{mit}$ accumulates with shorter stimulus cycle length (Brandes & Bers, 2002; Jo et al., 2006; Lu et al., 2013; Maack et al., 2006; Mason et al., 2020; Wust et al., 2017).

Extensive simulation analyses of systematically varying the distributions of $Ca^{2+}_{mit}$ and $Ca^{2+}_{SR}$ handling proteins revealed that the fraction of CaUni_js greatly affects the extent of $Ca^{2+}_{mit}$ accumulation during exercise; i.e. $Ca^{2+}_{mit}$ concentration increased 6.76–8.73 times assuming all CaUni faced the JS, while only 1.57–1.96 times assuming all CaUni faced the cytoplasm (Fig. 12). The role of CaUni_js in efficient $Ca^{2+}_{mit}$ accumulation is well comparable to the model estimation by Maack et al. (2006) showing that the introduction of a hypothetical extra-mitochondrial $Ca^{2+}$ microdomain and increasing the number and/or amplitude of $Ca^{2+}$ increase pulses effectively accumulate $Ca^{2+}_{mit}$. Interestingly, a larger fraction of NmSC resulted in a slight increase in $Ca^{2+}_{mit}$ for each given fraction of CaUni_js (Fig. 12*A*). This is explained as follows. Since NmSC supplies $Ca^{2+}$ from mitochondria to SR, a larger NmSC fraction facilitates $Ca^{2+}$ movement from mitochondria, increasing $Ca^{2+}_{SR}$ (Fig. 12*C*). Upon stimulation, more $Ca^{2+}$ is hence released from the SR via RyR, and CaUni_js flux is then increased and $Ca^{2+}_{mit}$ replenished. Thus, the overall contribution of the uneven distribution of $NCX_{mit}$ is additive to that of CaUni.

Although *in vitro* experiments clearly demonstrated that $Ca^{2+}$ activates NADH-producing dehydrogenases, it has been a matter of controversy whether $Ca^{2+}$-dependent regulation is required for metabolite constancy under *in vivo* physiological conditions (see reviews by Aon & Cortassa, 2012; Beard & Kushmerick, 2009; Glancy & Balaban, 2012; Korzeniewski, 2017; Saks et al., 2006;

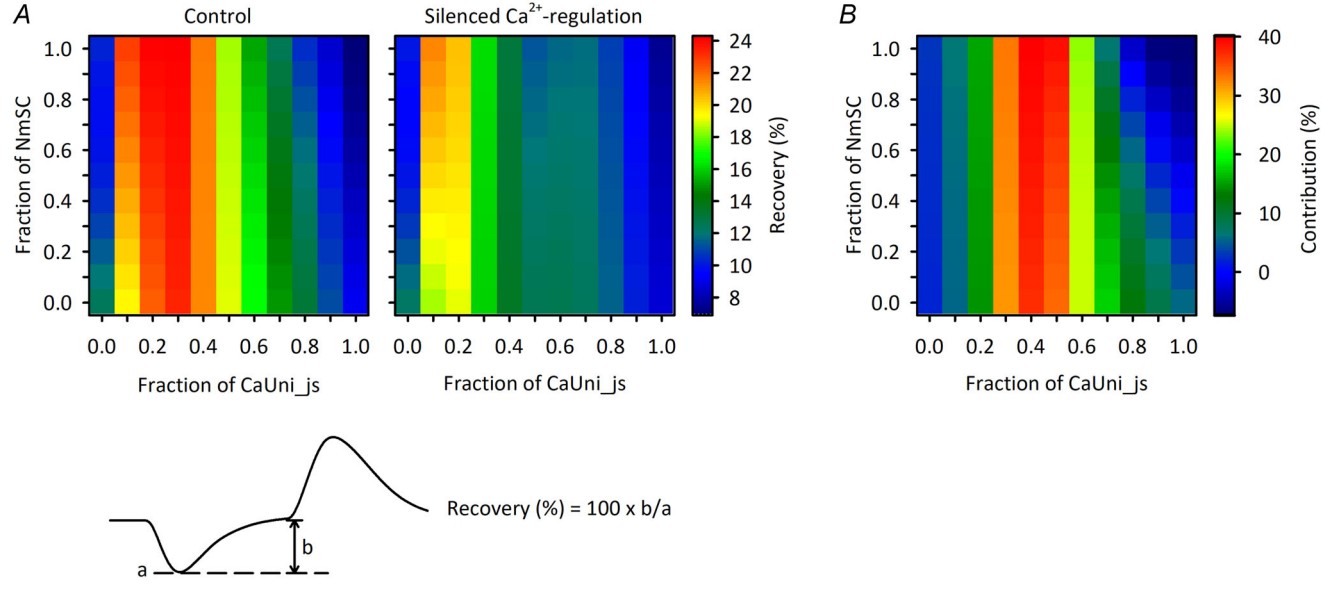

**Figure 13. Contribution of $Ca^{2+}_{mit}$-dependent regulation of dehydrogenases to $NADH_{mit}$ dynamics**
The fractional ratios of CaUni_js and NmSC were systematically varied as Fig. 12 and the exercise protocol in Fig. 7*A* was applied. *A*) recovery of $NADH_{mit}$ from initial drop without (left) and with (right) silenced $Ca^{2+}_{mit}$-dependent regulation. Recovery of $NADH_{mit}$ was calculated as $100 \times b/a$ (see lower panel). *B*) contribution of $Ca^{2+}_{mit}$-dependent regulation of dehydrogenases. The contribution (%) was calculated as; Contribution (%) = $\frac{(\text{Recovery (\%) in "Control"} - \text{Recovery (\%) with "silenced Ca}^{2+}\text{- regulation")}}{\text{Recovery (\%) in "Control"}} \times 100$.

Table 1. Energetics-related parameters of the Integrated Human Ventricular Cell Model with (MSI) or without (non-MSI) mitochondria-SR interaction.

| | MSI model | | | non-MSI model |
| --- | --- | --- | --- | --- |
| | 1 Hz | 2 Hz | 3 Hz | 1 Hz |
| ATPuse by contraction (mM/min) (% of total ATPuse) | 6.076 (62.99) | 10.935 (59.27) | 15.210 (57.49) | 3.886 (60.86) |
| ATPuse by SR Ca²⁺ uptake (mM/min) (% of total ATPuse) | 2.758 (28.59) | 6.331 (34.32) | 9.730 (36.78) | 1.667 (26.10) |
| ATPuse by $I_{Nak}$ (mM/min) (% of total ATPuse) | 0.748 (7.75) | 1.079 (5.85) | 1.374 (5.20) | 0.766 (12.00) |
| ATPuse by $I_{pCa}$ (mM/min) (% of total ATPuse) | 0.064 (0.67) | 0.104 (0.56) | 0.141 (0.53) | 0.067 (1.05) |
| total $ATP_{cyt}$ (mM) | 6.682 | 6.677 | 6.671 | 6.683 |
| total $ADP_{cyt}$ (mM) | 0.018 | 0.022 | 0.028 | 0.017 |
| $PCr_{cyt}$ (mM) | 15.556 | 14.257 | 12.857 | 16.031 |
| $Pi_{cyt}$ (mM) | 0.366 | 0.865 | 1.630 | 0.239 |
| PCr/ATP | 2.33 | 2.14 | 1.93 | 2.40 |
| $mVO_2$ (mM/min) | 1.882 | 3.431 | 4.880 | 1.273 |
| $\Delta\psi$ (mV) | −180.490 | −176.221 | −172.700 | −183.507 |
| $NADH_{mit}$ (mM) | 1.188 | 1.232 | 1.273 | 1.416 |

Takeuchi & Matsuoka, 2020). In fact, mice lacking MCU exhibited nearly normal cardiac function, including energy metabolism, with the exception of abrupt cardiac responses to catecholamine stimulation (Kwong et al., 2015; Luongo et al., 2015; Wu et al., 2015). We previously demonstrated, using a detailed model of mitochondria connected with simplified ATP consumption, that the contribution of $Ca^{2+}$ to steady state metabolite constancy at various workloads is small when the physiological metabolic substrate composition is set as for *in vivo* measurements (Saito et al., 2016). In the present Integrated Human Ventricular Cell Model, the $Ca^{2+}_{mit}$-dependent regulation accelerated $NADH_{mit}$ recovery from the initial drop during exercise, with a contribution of about 36.9%. The quasi-steady state $NADH_{mit}$ during the exercise protocol was maintained in a narrow range of −4.2 to +5.3% of the resting level (SD = 2.6%) in the control MSI model, and the variation was increased to 0 to −14.7% (SD = 4.0%) when $Ca^{2+}_{mit}$-dependent regulation of dehydrogenases was silenced. The decrease in SD suggested that the contribution of the $Ca^{2+}_{mit}$-dependent regulation is 35% ((4.0 − 2.6)/4.0 × 100). Accordingly, $NADH_{mit}$ became more stable in the presence of the regulation (Figs 7 and 9). The model predicted that $Ca^{2+}_{mit}$-dependent regulation would be effective only when $Ca^{2+}_{mit}$ increased to a certain level, i.e. > 400 nM (Fig. 9). This is in line with experimental findings that MCU knockout phenotypes became apparent only when cells were challenged with catecholamine (Kwong et al., 2015; Luongo et al., 2015; Wu et al., 2015).

As mentioned above, $NADH_{mit}$ change during exercise was kept relatively small even in the absence of $Ca^{2+}_{mit}$-dependent activation of dehydrogenases; see almost constant $NADH_{mit}$, especially in the range of 2–5 mM/min $mVO_2$ (Fig. 9E). What contributes to the stability of $NADH_{mit}$? One candidate regulator is Pi, an ATP hydrolysis product. Although early experiments reported a constant Pi concentration during workload transition (Katz et al., 1989), Wu et al. (2008) succeeded in detecting the change in Pi concentration during workload transition in *in vivo* canine hearts using $^{31}P$ MRS combined with model simulations. They estimated that the Pi concentration was 0.29 mM at baseline and reached 2.3 mM during workload increase. Our model analyses yielded comparable but slightly higher $Pi_{cyt}$ change; i.e. 0.37 mM at baseline, reaching 3.05 mM during exercise. This Pi entered the mitochondria, and allosterically activated OGDH, as reported experimentally (Rodriguez-Zavala et al., 2000), stabilizing $NADH_{mit}$ levels. It should be noted that allosteric activation by $Pi_{mit}$ of Complex III of the electron transport chain, which was considered in our previous model (Saito et al., 2016) but was subsequently refuted experimentally (Bazil et al., 2016; Vinnakota, Bazil et al., 2016), has been removed

in the present model. Another mechanism for NADH$_{mit}$ stability is feedback regulation of dehydrogenases by the substrate NADH/product NAD$^+$. As the substrate NADH is consumed and the product NAD$^+$ increases, then the fluxes of dehydrogenases should become slowed, preventing further decrease in NADH. In any case, the NADH$_{mit}$ stability should not be dependent solely on one regulatory mechanism, but on multiplex mechanisms.

In the present simulation analysis of exercise, mVO$_2$ increased 3.69-fold at maximum, which was comparable to values in the literature in humans (Binak et al., 1967; Heiss et al., 1976). At this exercise intensity, ATP$_{cyt}$ concentration remained almost constant at the expense of PCr$_{cyt}$ decrease, resulting in a decline of PCr/ATP (see Fig. 9$D$). Note that the extent of PCr/ATP decrease during standard exercise, 29.4%, was larger than the model-estimate by Bakermans et al. (2017), 10%, probably due to the higher workload in our model; mVO$_2$ was 6.31 mM/min in our model *vs.* 5 mM/min in Bakermans et al. (2017). The ATP$_{cyt}$ concentration and the extent of the decrease in PCr$_{cyt}$ and increases in Pi$_{cyt}$ and total ADP$_{cyt}$, and thus the decrease in PCr/ATP, were all unaffected by silencing Ca$^{2+}$$_{mit}$-dependent regulation (Fig. 7$F, G$ and Fig. 9$C, D, G, H$). These results suggest that Ca$^{2+}$$_{mit}$-dependent regulation has negligible effects on phosphate metabolites and ATP hydrolysis potential, and that adaptation of ATP supply to increased demand may proceed even in the absence of NADH$_{mit}$ change. The hypothesis is comparable to the report by Tran et al. (2015), who demonstrated using an excitation-contraction-energetics coupling model with simplified mitochondrial function that introducing Ca$^{2+}$$_{mit}$-dependent activation of dehydrogenase increased NADH$_{mit}$, yet did not affect ATP production. The contribution of Ca$^{2+}$$_{mit}$-dependent activation of dehydrogenases to ATP production should become apparent under more extreme physiological and pathophysiological conditions accompanying excessive ATP utilization, such as tachycardia and ventricular fibrillation (Badeer & Feisal, 1965; Kusuoka et al., 1992). Given our previous findings that metabolic substrate composition is an important determinant of the contribution of Ca$^{2+}$$_{mit}$-dependent activation of dehydrogenases (Saito et al., 2016), conditions accompanying metabolic perturbations would also be candidate situations when the Ca$^{2+}$$_{mit}$-dependent activation of dehydrogenases plays a more significant role. In fact, there are several inherited and acquired disorders associated with deficiencies in mitochondrial enzymes and therefore with metabolite remodelling (see reviews by Lopaschuk et al., 2021; Pell et al., 2016; Stanley et al., 2005). We tentatively focused on MDH, whose genetic mutations were recently reported (Priestley et al., 2022). When

exercise simulation was performed with MDH amplitude reduced to 0.1%, the extent of exercise-induced PCr$_{cyt}$ reduction was 43.3%, which was greater than 29.6% for 100% MDH. Interestingly, when the Ca$^{2+}$$_{mit}$-dependent regulation of dehydrogenases was silenced, the PCr$_{cyt}$ reduction became even greater, 55.2%. Similar results were obtained when all metabolic substrates concentrations were reduced to 15%; i.e. the extent of PCr$_{cyt}$ reduction by exercise was 39.9% and 53.6% in the presence and absence of the Ca$^{2+}$$_{mit}$-dependent regulation. It would be interesting to evaluate excitation-contraction-mitochondrial energetics coupling by considering the remodelling of Ca$^{2+}$ handling proteins and energetics-related proteins as well as of metabolite composition. However, these are beyond the scope of the present study.

## Limitation of the model

Our model has several limitations. First, we compared our model analyses data with human *in vivo* experimental data. Since the model is of a single ventricular myocyte, there are some deviations. For example, the mVO$_2$ value calculated form the model, 1.88 mM/min at stimulation cycle length of 1 s, is approximately half the value estimated from human data; i.e. 3.57 mM/min (calculated from ∼8 ml/100 g/min at rest (Strauer, 1979)) and can increase by ∼10-fold at maximal exercise (see also textbook by Herring & Paterson, 2018). Factors which were not included in this single ventricular myocyte model such as afterload, oxygen delivery, involvements of atria, and so on, should affect the results. Most recently, Sturgess et al. (2024) reported mechanistic insights into myocardial perfusion and oxygen delivery during exercise, using a sophisticated model that integrates whole-body cardiovascular haemodynamics, cardiac mechanics and myocardial work. Our integrated cardiomyocyte model analyses in combination with whole-body modelling analyses would provide more accurate and interesting information, albeit at an enormous computational cost.

Second, we assume an extremely strong coupling between mitochondria and SR, as was done in the HL-1 cell model, since the details of coupling mechanisms remain unresolved (Takeuchi & Matsuoka, 2022). The contribution of NmSC to Ca$^{2+}$$_{mit}$ and Ca$^{2+}$$_{SR}$ dynamics might be overestimated in the model.

Third, the present model calculates the ATP consuming steps according to the reported stoichiometry and does not consider the ATP-dependence of SERCA, $I_{NaK}$, $I_{pCa}$, and myosin ATPase. In addition to this, we did not consider the ATP-sensitive K$^+$ channels at the plasma membrane and mitochondria that open upon ATP

starvation (Noma, 1983; see also review Foster & Coetzee, 2016), making it impossible to simulate ATP-starved conditions. Introduction of thermodynamically consistent models for ATP-consuming components (Pan et al., 2019; Tran et al., 2010), as well as models for ATP-sensitive $K^+$ channels (Himeno et al., 2024; Zhou et al., 2009), and testing the ATP-starving conditions such as ischemia-reperfusion, would provide further insights into excitation-contraction-mitochondrial energetics coupling.

Fourth, the present model does not include glycolysis and $\beta$-oxidation of fatty acid. However, the absence of these processes is unlikely to have much influence on the main conclusion. When the $Ca^{2+}_{mit}$-dependent activation of ICDH and OGDH was silenced but that of PDHC was intact during the exercise simulation, the time course of $NADH_{mit}$ almost overlapped with the dashed grey line in the Fig. 7*A*; the recovery of $NADH_{mit}$ from its initial drop was 12% in both simulations. This is mainly due to the relatively small $Ca^{2+}$-sensitive component of PDHC (Saito et al., 2016). Therefore, the potential $Ca^{2+}_{mit}$ effects would not be greatly affected even when the PDHC contribution is changed by incorporating glycolysis and $\beta$-oxidation of fatty acid. However, substrate usage of mitochondria is an important issue. It would be interesting to examine the effects of glucose-fuelled *versus* fatty acid-fuelled respiration on cardiac energetics during workload transition, but it is beyond the scope of the present study.

**Figure A1. Dependence of mVO$_2$ and NADH$_{mit}$ on $Ca^{2+}_{cyt}$ and $Pi_{cyt}$ in the isolated mitochondrial model**
*A*) dependence of $Ca^{2+}_{cyt}$. The isolated mitochondrial model was calculated for 1 h with $Ca^{2+}_{cyt}$ of $10^{-9} - 2.0 \times 10^{-3}$ mM. The steady state values were obtained and normalized to those at $10^{-9}$ mM $Ca^{2+}_{cyt}$ (filled circles). Experimental data using isolated porcine cardiac mitochondria were from Territo et al. (2000). *B*) dependence of $Pi_{cyt}$. The isolated mitochondrial model was serially applied with $Pi_{cyt}$ of $10^{-6} - 10$ mM and $ADP_{cyt}$ of 1.3 mM at 110 s and 170 s, respectively. The values at 169 s (state IV) and 229 s (state III) were obtained. For $NADH_{mit}$, the values were expressed as the percentage of the reduced form, i.e. $NADH_{mit}$, to total $NAD^+_{mit}$ plus $NADH_{mit}$ (filled triangles). For mVO$_2$, the values were normalized to those at $10^{-6}$ mM $Pi_{cyt}$ (filled circles). Experimental data using isolated porcine cardiac mitochondria were from Bose et al. (2003).

## Appendix

Figures A1–A5

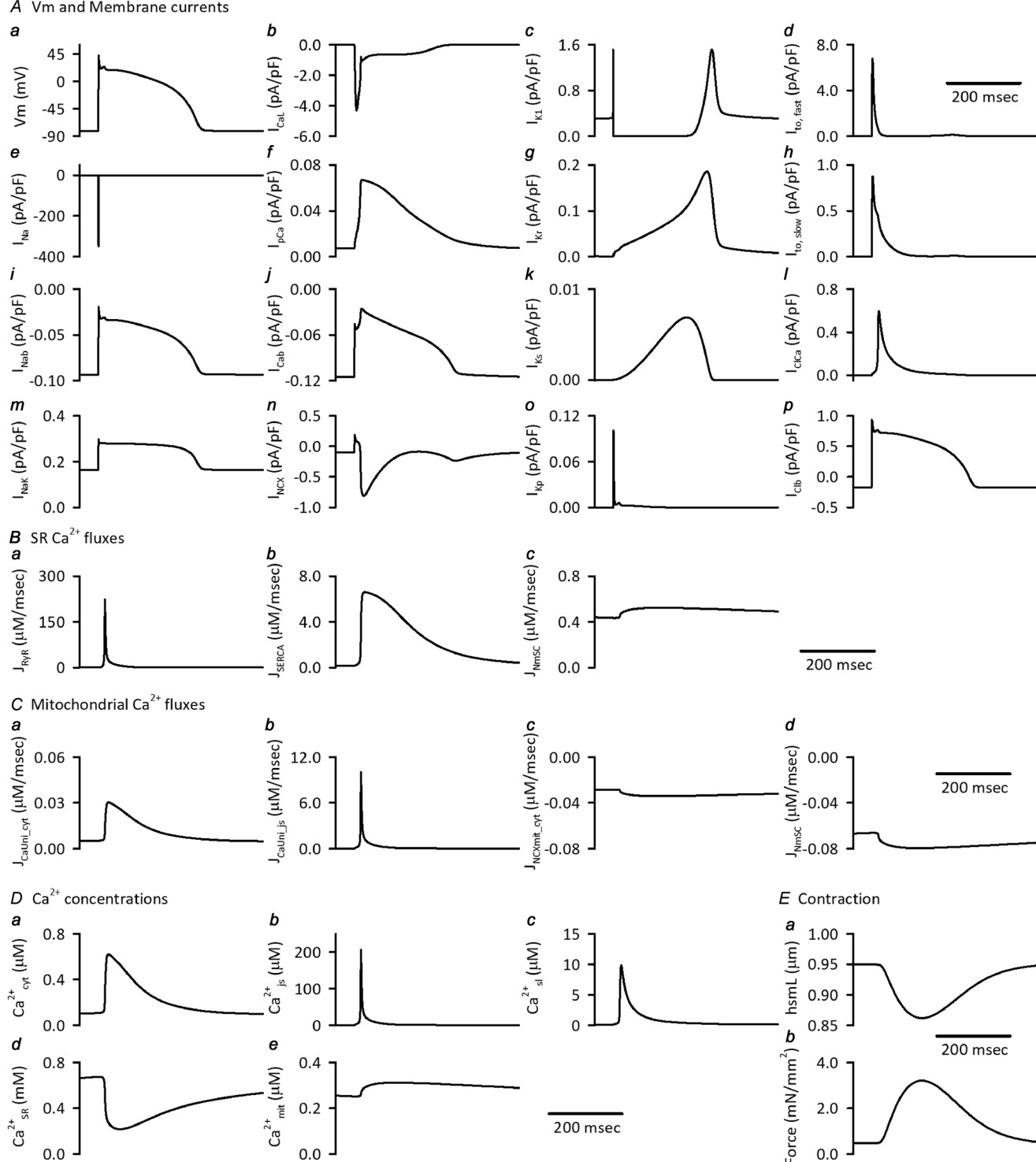

**Figure A2. Configurations of action potential, major ionic currents, Ca²⁺ fluxes and Ca²⁺ concentrations in each compartment of the Integrated Human Ventricular Cell Model**
The standard initial condition of the model was obtained without applying $F_{ext}$, i.e. 0.950 $\mu$m diastolic hsmL, under the condition of isotonic contraction, stimulated with a cycle length of 1 s for 30 min. The steady state values are listed in the Online Supplementary Material Table S14.

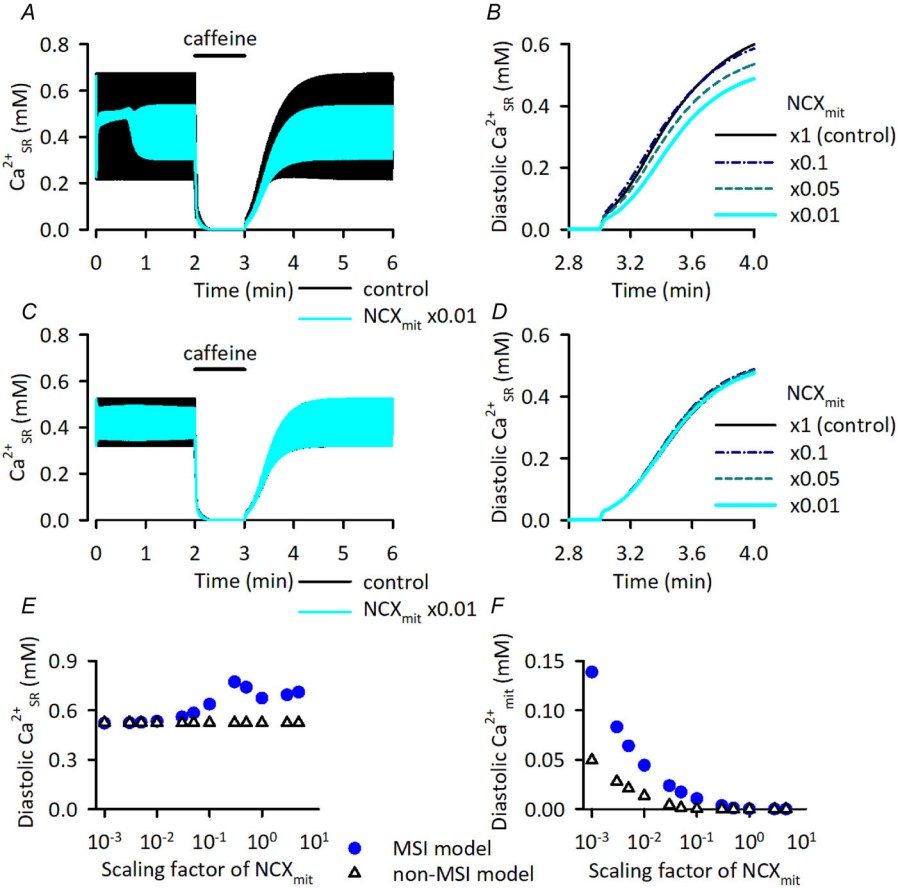

**Figure A3. Effects of NCX$_{mit}$ reduction on the responses to caffeine application and removal of the Integrated Human Ventricular Cell Model with and without mitochondria-SR interaction**

*A*) time courses of Ca$^{2+}$$_{SR}$ concentration in the model with MSI. Black and blue lines represent data using the model with NCX$_{mit}$ amplitude factor of ×1.0 (control) and ×0.01, respectively. *B*) same protocol was applied with various amplitude factors for NCX$_{mit}$. The Ca$^{2+}$$_{SR}$ reuptake phase is magnified. *C* and *D*) the same protocols were applied to the model without MSI (non-MSI), as in *A* and *B*, respectively. *E*) relationships between NCX$_{mit}$ amplitude factor and diastolic Ca$^{2+}$$_{SR}$ at the end of the protocol, in the model with (control; blue filled circles) and without (open triangles) MSI. *F*) relationships between NCX$_{mit}$ amplitude factor and diastolic Ca$^{2+}$$_{mit}$ at the end of the protocol, in the model with (control; blue filled circles) and without (open triangles) MSI.

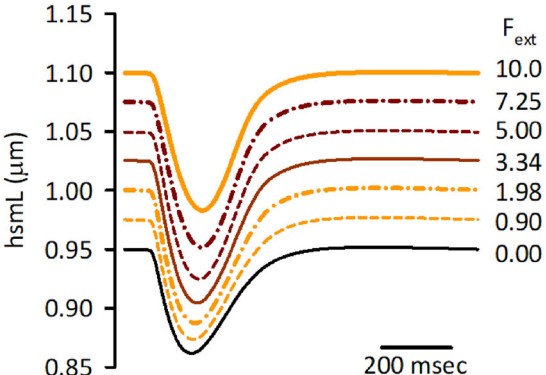

**Figure A4. The hsmL shortenings with different $F_{ext}$ in the Integrated Human Ventricular Cell Model**

The model cell was stimulated with a cycle length of 1 s with various $F_{ext}$ values of 0.00–10.00; i.e. 0.950–1.100 $\mu$m diastolic hsmL. The hsmL shortenings at steady state were plotted.

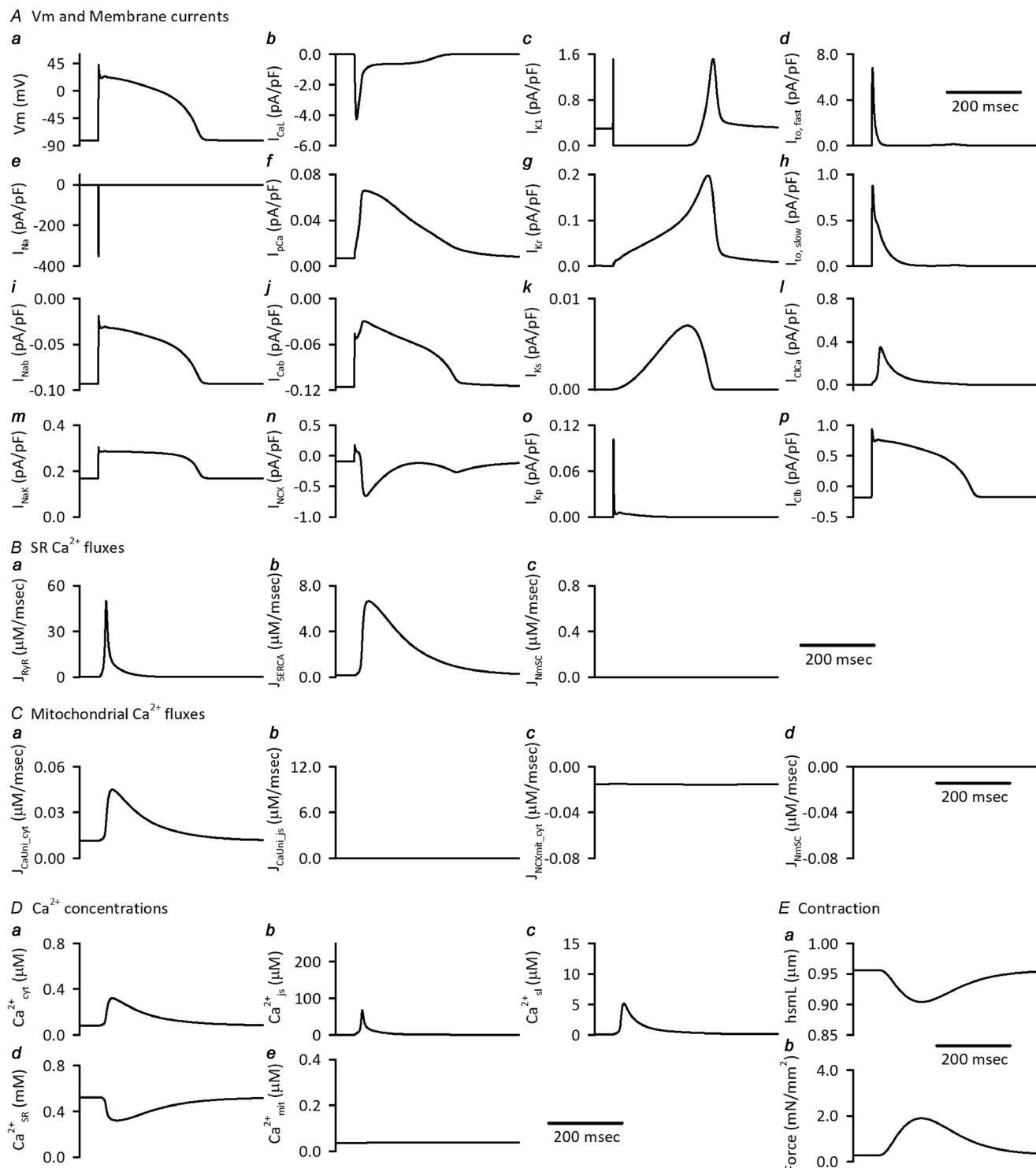

**Figure A5. Configurations of action potential, major ionic currents, Ca²⁺ fluxes and Ca²⁺ concentrations in each compartment of the Integrated Human Ventricular Cell Model without mitochondria-SR interaction (non-MSI)**

The Integrated Human Ventricular Cell Model without mitochondria-SR interaction (non-MSI) was obtained by changing the fractions of NmSC and CaUni_js to 0, and then calculated for 20 min using the standard initial conditions of the MSI model. The steady state values are shown in the Online Supplementary Material Table S14.

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

## Additional information

### Data availability statement

The source code is available on GitHub at https://github.com/atakeuti/IHVCM. The original contributions presented in the study are included in the article and in the Online Supplementary Material; further inquiries can be directed to the corresponding author.

### Competing interests

The authors declare that they have no conflicts of interest with the contents of this article.

### Author contributions

A.T.: conceptualization, study design, software, data analyses, visualization, writing the original draft. S.M.: conceptualization, study design, software, data analyses, visualization, review & editing the originals draft. Both authors approved the final version of the manuscript, and agreed to be accountable for all aspects of the work in ensuring that questions related to the accuracy or integrity of any part of the work are appropriately investigated and resolved, and confirmed that both persons designated as authors qualify for authorship and all those who qualify for authorship are listed.

### Funding

This work was supported by JSPS KAKENHI grant Numbers 18K06869 (A.T.), 22K06841 (A.T.), 19H03400 (S. M.), and 23K24065 (S. M.), by Life Science Innovation Centre at University of Fukui (A.T.), Research Grants from the University of Fukui (FY 2023) (A.T.), and by the Kurata Grants of The Hitachi Global Foundation (A.T., grant Number 1511).

### Keywords

energetics, exercise, heart, mathematical modelling, mitochondria, sarcoplasmic reticulum

### Supporting information

Additional supporting information can be found online in the Supporting Information section at the end of the HTML view of the article. Supporting information files available:

**Peer Review History**
**Online Supplementary Material**

