## [Peer Review History · The Journal of Physiology]

A simulation study on the role of mitochondria-sarcoplasmic reticulum Ca²⁺ interaction in cardiomyocyte energetics during exercise

Ayako Takeuchi and Satoshi Matsuoka

DOI: 10.1113/JP286054

Corresponding author(s): Ayako Takeuchi (atakeuti@u-fukui.ac.jp)

The following individual(s) involved in review of this submission have agreed to reveal their identity: Daniel A Beard (Referee #1); Kunichika Tsumoto (Referee #2)

Review Timeline:

Submission Date:	16-Jun-2024
Editorial Decision:	01-Aug-2024
Revision Received:	08-Aug-2024
Accepted:	15-Aug-2024

Senior Editor: Vaughan Macefield

Reviewing Editor: Yoshihiro Kubo

Transaction Report:

Dear Dr Takeuchi,

Re: JP-RP-2024-286054 "A simulation study on the role of mitochondria-sarcoplasmic reticulum Ca²⁺ interaction in cardiomyocyte energetics during exercise" by Ayako Takeuchi and Satoshi Matsuoka

Thank you for submitting your manuscript to The Journal of Physiology. It has been assessed by a Reviewing Editor and by 2 expert referees and we are pleased to tell you that it is acceptable for publication following satisfactory revision.

REVISION CHECKLIST:

- 'Potential Cover Art' for consideration as the issue's cover image
- Appropriate Supporting Information (Video, audio or data set: see https://jp.msubmit.net/cgi-bin/main.plex?form_type=display_requirements#supporting_information)

form_type=display_requirements#supp).

We look forward to receiving your revised submission.

Yours sincerely,

Vaughan Macefield
Senior Editor
The Journal of Physiology

REQUIRED ITEMS

- Author photo and profile. First or joint first authors are asked to provide a short biography (no more than 100 words for one author or 150 words in total for joint first authors) and a portrait photograph. These should be uploaded and clearly labelled together in a Word document with the revised version of the manuscript. See Information for Authors for further details.

- Your paper contains Supporting Information of a type that we no longer publish, including supplementary tables and figures. Any information essential to an understanding of the paper must be included as part of the main manuscript and figures. The only Supporting Information that we publish are video and audio, 3D structures, program codes and large data files. Your revised paper will be returned to you if it does not adhere to our Supporting Information Guidelines.

- Please include an Abstract Figure file, as well as the Figure Legend text within the main article file. The Abstract Figure is a piece of artwork designed to give readers an immediate understanding of the research and should summarise the main conclusions. If possible, the image should be easily 'readable' from left to right or top to bottom. It should show the physiological relevance of the manuscript so readers can assess the importance and content of its findings. Abstract Figures should not merely recapitulate other figures in the manuscript. Please try to keep the diagram as simple as possible and without superfluous information that may distract from the main conclusion(s). Abstract Figures must be provided by authors no later than the revised manuscript stage and should be uploaded as a separate file during online submission labelled as File Type 'Abstract Figure'. Please also ensure that you include the figure legend in the main article file. All Abstract Figures should be created using BioRender. Authors should use The Journal's premium BioRender account to export high-resolution images. Details on how to use and access the premium account are included as part of this email.

EDITOR COMMENTS

Reviewing Editor:

The authors previously reported critical roles of the MCU- RyR coupling and the NCLX-SERCA in the mitochondria - SR Ca²⁺ dynamics in cardiac myocytes, using computational and experimental approaches. They also achieved previously the integration of the metabolic factors to approach the energetic aspect of cardiac myocytes.

In the present study, they aimed to elucidate the contribution of Ca²⁺ dependent mechanisms to energetics and overall function of cardiomyocyte during dynamic changes of workload in exercise. By computational approaches, the authors performed integrated simulation analysis of excitation-contraction-energetics coupling in human ventricular myocytes. They observed that the coupling of Ca_{uni} and RyR is especially important for the Ca_{2+mit} level during exercise, and that it contributes to the regulation of NADH_{mit} level via Ca_{2+mit}-dependent NADH producing dehydrogenases. I highly evaluate the impact and scientific merits of the findings achieved by thorough analysis.

Both reviewers also highly evaluated this paper, and their comments are mostly minor. I trust that there will be no difficulty to revise the manuscript suitably in response to their comments.

I understand this is a commissioned article to a special issue entitled "Regulation of Cardiovascular and Skeletal Muscle Function in Exercise", and I believe this paper is truly suitable to the topics. I would like to thank the authors for their effort in response to our request to submit an article to the special issue.

Please also see 'Required Items' above.

Senior Editor:

Thank you for submitting your manuscript to The Journal of Physiology. I have now received comments from two independent reviewers and the Reviewing Editor, all experts in the field. As you will see from their comments, there are some issues you will need to address before we can consider the manuscript further. I invite you to revise the manuscript accordingly and submit point-by-point responses to the reviewers' comments. Please also include a statement regarding the source of the computer codes and whether this and the data can be shared. I look forward to receiving your revised manuscript in due course.

REFeree COMMENTS

Referee #1:

My apologies for being very late with this review. Any delays on a decision are the fault of this reviewer and not the editor! My only excuse is that this is a very long paper. In fact, it is a tour-de-force and I believe an important contribution. I have little to criticize and could recommend publication as is. I have a few points to raise--none of them crucial.

Line 106-108: It is pointed out that in prior models, when physiological changes in calcium are imposed there is little to no effect on NADH production. I might point out that the same thing has been seen experimentally in suspensions of isolated mitochondria: PMID: 21757763, PMID: 26910432.

Line 294: This is maybe an important point about ADP and Pi increasing and PCr decreasing in exercise. It is shown how much ADP and Pi are predicted to change, but not discussed how much PCr is predicted to change. (It is a very good approximation that PCr decreases as much as Pi increases.) Bakermans et al. (<https://doi.org/10.3389/fphys.2017.00939>) predict that PCr does not change enough to be observable. Are the current prediction in line with that analysis?

The comparison between Figure 9 panels C and F is not discussed. This seems important to me. My interpretation is that Ca is predicted to have a (secondary?) effect on steady-state NADH levels, but seems to have no predicted effect on energetics in terms of phosphate metabolites and ATP hydrolysis potential. What are your thoughts?

I think that the model implicitly assumes carbohydrate substrate, with reliance on PDH. I would speculate that with fatty acid fueled respiration the potential calcium effects would be diminished because the calcium-dependence of PDH is bypassed.

Referee #2:

The energy expended in cardiomyocyte contraction is considered one of the most fundamental quantities in cardiac mechanics. Even though, no experimental and assay systems have yet been established to gain physiological insight.

In the present study, Takeuchi and Matsuoka examine cardiac energy dynamics during exercise loading by constructing and analyzing a sophisticated cellular model with excitation-contraction-energetics coupling. This study builds on previous work by this group that formulated a mitochondrial Ca²⁺-dependent energy metabolism regulatory mechanism and recapitulated SR Ca²⁺ dynamics by considering Ca²⁺ interactions between mitochondria and the sarcoplasmic reticulum (SR) and proposes an extremely sophisticated new integrated human ventricular myocyte model that integrates the SR-mitochondrial Ca²⁺ dynamics with the human ventricular myocyte model (here based on the Grandi-Bers 2011 model). They quantified

cardiac energy dynamics during dynamic load transitions during exercise and the contribution of Ca²⁺-dependent regulatory mechanisms controlling mitochondrial energy metabolism to cardiomyocyte function.

The strengths of this study are the detailed analysis of changes in all factors of heart rate, preload, and contractility during dynamic load transitions during exercise, or the contribution to cardiac energy dynamics associated with these individual changes, and the systematic analysis of the effects of the presence or absence of Ca²⁺-dependent energy metabolism regulatory mechanisms within mitochondria on cardiomyocyte function. In particular, an important role in the spatial interaction between mitochondrial Ca²⁺ uniporters and ryanodine receptors was identified, demonstrating that Ca²⁺-dependent activation of NADH-producing dehydrogenases within mitochondria contributes to NADH stabilization during exercise. Such superb sophisticated and integrated models from the standpoint of systems physiology have the potential to establish a new assay system using in silico systems.

The manuscript is detailed and well-described. Hence, this reviewer will only point out a few errors in the manuscript and its presentation.

Please see the following:

1. Please unify the abbreviations for mitochondrial Ca²⁺ uniporter; MCU and CaUni are mixed in the manuscript.
2. P.7, Lines 148-149: The mitochondria-SR interaction (MSI) component has been developed based on experimental observations using HL-1 cells (based on Takeuchi & Matsuoka, 2022). However, in integrating it into the human cardiomyocyte model, is there any part of the HL-1 cell model that has been adjusted to fit the human cardiomyocyte model? This should be mentioned if adjustments have been made based on in vitro experimental data using cardiomyocytes in humans.
3. p.8, Lines 159-160: "100/70 times" This expression is different from the rest of the document, is there a reason for this?
4. p.9, Line 199: I wonder if the result of Pi vs NADHmit presented at the bottom of Appendix Fig. A1B does not reproduce the experimental data well. Any comments?
5. p.12, Line 254: given the statement in p.11, Lines 239-240, is "0.3" a mistake for 0.35? And then Pca,0, Pna,0, and PK,0 need to be explained.
6. P.12, Line 255: Similar to #5, "0.3" could be a mistake for 0.35. Further explanation of Vmax_SERCA,0 is also needed.
7. P.12, Line 256: Vmax_INaK,0 explanation needed.
8. P.12, Line 257: Explanation of GKs,0 needed.
9. P.13, Lines 264-266: Shouldn't authors comment on the correspondence with the original formula (supplementary materials)?
10. Essentially, an explanation of each symbol in equations on p. 12 and p. 13 is needed. It may be complicated.
11. P.15, Lines 318-319: Looking at the left panel of Figures 6c and 6d, there is indeed an increase in flux. However, although it is stated in the main text that both consumption and production fluxes are increasing, they do not appear to be depicted together. Thus, the flux difference plots do not reveal what the difference is.
12. Fig.6 Legend: it lacks a brief description of panels C and D.
13. P.17, Lines 363-364: I couldn't understand where this sentence refers to. Does the flux in Complex I refer to Fig. 8A?
14. P.18, Line 372: "0.83-0.33 sec"; On the other hand, the figure legend of Fig. 9 says the CL shortening of 1.0-0.33 sec. Which is correct? In addition, shouldn't the CL shortening cases be clearly stated? For example, 1.0, 0.83, 0.5, 0.33, etc.
15. P.18, Line 388: "Ca²⁺mi" Typo? Correct "Ca²⁺mit"?
16. 32. p.19, Lines 411-412: "However, ... diminished." Where should we look for the results in this sentence? The Ca²⁺cyt transient, the decrease in Ca²⁺mit, and the decrease in Ca²⁺ flux via RyR can be understood from Fig. A5 (but I think it should be clearly stated to be compared to Fig. 2A). On the other hand, where should we look for hsmL shortening and decreased ATP consumption.

17. p.19, Lines 413-414: "Accordingly, ... MSI model." Where is this sentence attributed from?

18. P.22, Line 493: "CaUni_t"; is this typo? CaUni_cyt or CaUni?

19. P.57, Figure 9: The plot is a little difficult to read, so please revise it.

END OF COMMENTS

Confidential Review

16-Jun-2024

August 8th, 2024

Professor Vaughan Macefield
Senior Editor
The Journal of Physiology

Revision of JP-RP-2024-286054

Thank you for your positive reply. We have carefully studied the comments by Editors and Referees, and revised manuscript according to their suggestions. We replied point-by-point to the comments as coloured in blue below. In the manuscript, we highlighted in blue any changes, including typos and errors. Figures 6, 9, A1, A2 and A5 were updated according to the referees' comments. In addition, we updated Figure 1A by specifying the place "T-tubule" to improve clarity. We also updated our "Online Supplementary Material", in which unit descriptions were unified.

Sincerely yours,
Ayako Takeuchi

REQUIRED ITEMS:

- *Author photo and profile. First or joint first authors are asked to provide a short biography (no more than 100 words for one author or 150 words in total for joint first authors) and a portrait photograph. These should be uploaded and clearly labelled together in a Word document with the revised version of the manuscript. See Information for Authors for further details.*

A short biography and a portrait photograph of the first author were uploaded.

- *Your paper contains Supporting Information of a type that we no longer publish, including supplementary tables and figures. Any information essential to an understanding of the paper must be included as part of the main manuscript and figures. The only Supporting Information that we publish are video and audio, 3D structures, program codes and large data files. Your revised paper will be returned to you if it does not adhere to our Supporting Information Guidelines.*

Any information essential to an understanding of the paper is included as part of the main manuscript and figures. In addition, our "Online Supplementary Material" contains all equations and parameters of the model, and is 46 pages long, which we consider too large to be shown in the Appendix. Furthermore, although we have made the source code public, a description of all the equations should be useful to many readers who are not familiar with programming. Therefore, we decided to submit it as Supporting Information.

- *Please include an Abstract Figure file, as well as the Figure Legend text within the main article file. The Abstract Figure is a piece of artwork designed to give readers an immediate understanding of the research and should summarise the main conclusions. If possible, the image should be easily 'readable' from left to right or top to bottom. It should show the physiological relevance of the manuscript so readers can assess the importance and content of its findings. Abstract Figures should not merely recapitulate other figures in the manuscript. Please try to keep the diagram as simple as possible and without superfluous information that may distract from the main conclusion(s). Abstract Figures must be provided by authors no later than the revised manuscript stage and should be uploaded as a separate file during online submission labelled as File Type 'Abstract Figure'. Please also ensure that you include the figure legend in the main article file. All Abstract Figures should be created using BioRender. Authors should use The Journal's premium BioRender account to export high-resolution images. Details on how to use and access the premium account are included as part of this email.*

We created and uploaded an Abstract Figure. We also included the Abstract Figure legend in the main article file (p43).

EDITOR COMMENTS

REVIEWING EDITOR COMMENTS:

The authors previously reported critical roles of the MCU-RyR coupling and the NCLX-SERCA in the mitochondria - SR Ca²⁺ dynamics in cardiac myocytes, using computational and experimental approaches. They also achieved previously the integration of the metabolic factors to approach the energetic aspect of cardiac myocytes.

In the present study, they aimed to elucidate the contribution of Ca²⁺ dependent mechanisms to energetics and overall function of cardiomyocyte during dynamic changes of workload in exercise. By computational approaches, the authors performed integrated simulation analysis of excitation-contraction-energetics coupling in human ventricular myocytes. They observed that the coupling of CaUni and RyR is especially important for the Ca²⁺mit level during exercise, and that it contributes to the regulation of NADHmit level via Ca²⁺mit-dependent NADH producing dehydrogenases. I highly evaluate the impact and scientific merits of the findings achieved by thorough analysis.

Both reviewers also highly evaluated this paper, and their comments are mostly minor. I trust that there will be no difficulty to revise the manuscript suitably in response to their comments.

I understand this is a commissioned article to a special issue entitled "Regulation of Cardiovascular and Skeletal Muscle Function in Exercise", and I believe this paper is truly suitable to the topics. I would like to thank the authors for their effort in response to our request to submit an article to the special issue.

Please also see 'Required Items' above.

Thank you for your high evaluation on our work. We have carefully studied the comments by Editors and Referees, and revised manuscript according to their suggestions. We replied point-by-point to the comments as coloured in blue below.

SENIOR EDITOR COMMENTS:

Thank you for submitting your manuscript to The Journal of Physiology. I have now received comments from two independent reviewers and the Reviewing Editor, all experts in the field. As you will see from their comments, there are some issues you will need to address before we can

consider the manuscript further. I invite you to revise the manuscript accordingly and submit point-by-point responses to the reviewers' comments. Please also include a statement regarding the source of the computer codes and whether this and the data can be shared. I look forward to receiving your revised manuscript in due course.

Thank you for your high evaluation on our work. We have carefully studied the comments by Editors and Referees, and revised manuscript according to their suggestions. We replied point-by-point to the comments as coloured in blue below. In addition, we made the source code available on GitHub, and included the statement in the manuscript as follows;

“The source code is available on GitHub at <https://github.com//atakeuti/IHVCM>” (p6, lines 19-20; and p39, “Data availability statement” section).

REFEREE COMMENTS:

Referee #1:

My apologies for being very late with this review. Any delays on a decision are the fault of this reviewer and not the editor! My only excuse is that this is a very long paper. In fact, it is a tour-de-force and I believe an important contribution. I have little to criticize and could recommend publication as is. I have a few points to raise--none of them crucial.

Thank you for your high evaluation on our work. We have revised the manuscript according to the comments.

Line 106-108: It is pointed out that in prior models, when physiological changes in calcium are imposed there is little to no effect on NADH production. I might point out that the same thing has been seen experimentally in suspensions of isolated mitochondria: PMID: 21757763, PMID: 26910432.

Thank you for the comment. We added description regarding experimental findings as follows;

“which were in line with experimental reports using isolated rat cardiac mitochondria (Vinnakota et al., 2011, 2016a).” (p6, lines 1-3).

Accordingly, we added the two literatures as references and renumbered the Vinnakota et al., 2016 in the original version to Vinnakota et al., 2016b (p9, line 15; p25, line 18).

Line 294: This is maybe an important point about ADP and Pi increasing and PCr decreasing in exercise. It is shown how much ADP and Pi are predicted to change, but not discussed how much PCr is predicted to change. (It is a very good approximation that PCr decreases as much

as Pi increases.) Bakermans et al. (<https://doi.org/10.3389/fphys.2017.00939>) predict that PCr does not change enough to be observable. Are the current prediction in line with that analysis?

Thank you for the valuable comment. According to the presentation by Bakermans et al., 2017, we added new panels for the heart rate-PCr/ATP relationships in Figure 9 (Figure 9D and H). The PCr/ATP reduction was 17% when heart rate was increased from 60 to 180 min⁻¹, and 29.4% in our standard exercise condition (cycle length shortening from 1.00 to 0.33 sec, i.e., heart rate increase from 60 to 180 min⁻¹, F_{ext} increase from 0.0 to 5.0, and $\alpha=1.0$ for β -adrenergic stimulation) regardless of the existence of the Ca²⁺_{mit}-dependent mechanisms. The PCr/ATP reduction values were larger than the model-estimate by Bakermans et al., 2017, 10%. We describe and discuss in the “Results” and “Discussion” sections as follows;

“On the other hand, silencing the Ca²⁺_{mit}-dependent regulation did not alter ADP_{cyt}, Pi_{cyt} and PCr/ATP (Figure 9C vs G, D vs H).” (p20, lines 13-14)

“At this exercise intensity, ATP_{cyt} concentration remained almost constant at the expense of PCr_{cyt} decrease, resulting in a decline of PCr/ATP (see Figure 9D). Note that the extent of PCr/ATP decrease by standard exercise, 29.4%, was larger than the model-estimate by Bakermans et al. 2017, 10%, probably due to the higher workload in our model; the mVO₂ is 6.31 vs 5 mM/min in our vs in Bakermans et al., 2017, respectively.” (p26, lines 1-5).

Accordingly, we re-numbered panels D-F in the original version of Figure 9 to panels E-G, revised legends of Figure 9 (p47), and added the Bakermans et al., 2017 as a reference.

Related to this, we added the phrases in the “Discussion” section to improve clarity underlined as follows;

“Wu et al. (2008) succeeded in detecting the change in Pi concentration during workload transition in *in vivo* canine hearts using ³¹P MRS combined with model simulations.” (p25, lines 10-11).

“Our model analyses yielded comparable but slightly higher Pi_{cyt} change;” (p25, line 13).

The comparison between Figure 9 panels C and F is not discussed. This seems important to me. My interpretation is that Ca is predicted to have a (secondary?) effect on steady-state NADH levels, but seems to have no predicted effect on energetics in terms of phosphate metabolites and ATP hydrolysis potential. What are your thoughts?

Thank you for the comment. We agree with the referee that a comparison of panels C and G (formerly F) in the Figure 9 suggests that Ca^{2+} has no effect on energetics in terms of phosphate metabolites and ATP hydrolysis potential. We added the description in the “Discussion” section as follows;

“The ATP_{cyt} concentration and the extents of PCr_{cyt} decrease, Pi_{cyt} and total ADP_{cyt} increases, and thus PCr/ATP decrease, were all unaffected by silencing $\text{Ca}^{2+}_{\text{mit}}$ -dependent regulation (Figure 7F, G, and Figure 9C, D, G, H). These results suggest that $\text{Ca}^{2+}_{\text{mit}}$ -dependent regulation has negligible effects on phosphate metabolites and ATP hydrolysis potential, and that.....” (p26, lines 5-9).

In addition, to help readers easily compare “Control” and “Silenced Ca^{2+} -regulation”, we enlarged the symbols as well as the panels, and added grids to the panels in Figure 9.

I think that the model implicitly assumes carbohydrate substrate, with reliance on PDH. I would speculate that with fatty acid fueled respiration the potential calcium effects would be diminished because the calcium-dependence of PDH is bypassed.

Thank you for the comment. Prompted by the referee’s idea, we rescued $\text{Ca}^{2+}_{\text{mit}}$ -dependent activation of PDHC in the exercise simulation with silenced $\text{Ca}^{2+}_{\text{mit}}$ -dependent regulation, i.e., $\text{Ca}^{2+}_{\text{mit}}$ -dependent regulation was active in PDHC but inactive in ICDH and OGDH. As a result, the time course of NADH_{mit} was almost overlapped with that with silenced $\text{Ca}^{2+}_{\text{mit}}$ -dependent regulation in PDHC, ICDH, and OGDH; the recovery of NADH_{mit} from initial drop was 12% in both simulations. This is mainly due to the relatively small Ca^{2+} -sensitive component of PDHC (see Figure S16 in Saito et al., *J Physiol*, 2016, which was based on experimental data by Moreno-Sánchez and Hansfordort, *Biochem J*, 1988). Therefore, we believe that the potential Ca^{2+} effects would not be largely affected even when PDHC is bypassed. However, substrate usage of mitochondria is an important issue. It would be interesting to examine the effects of glucose-fuelled versus fatty acid-fuelled respiration on cardiac energetics during workload transition, but it is beyond the scope of the present study. We added the statements in the “Discussion” section as follows;

“Fourth, the present model does not include glycolysis and β -oxidation of fatty acid. However, the absence of these processes unlikely to have much influence on the main conclusion. When the $\text{Ca}^{2+}_{\text{mit}}$ -dependent activation of ICDH and OGDH was silenced but the one of PDHC was intact during the exercise simulation, the time course of NADH_{mit} was almost overlapped with the dashed grey line in the Figure 7A; the recovery of NADH_{mit} from initial drop was 12% in both simulations. This is mainly due to the

relatively small Ca^{2+} -sensitive component of PDHC (Saito et al., 2016). Therefore, the potential $\text{Ca}^{2+}_{\text{mit}}$ effects would not be largely affected even when PDHC contribution is changed by incorporating glycolysis and β -oxidation of fatty acid. However, substrate usage of mitochondria is an important issue. It would be interesting to examine the effects of glucose-fuelled versus fatty acid-fuelled respiration on cardiac energetics during workload transition, but it is beyond the scope of the present study.” (p28, lines 11-21)

Referee #2:

The energy expended in cardiomyocyte contraction is considered one of the most fundamental quantities in cardiac mechanics. Even though, no experimental and assay systems have yet been established to gain physiological insight.

In the present study, Takeuchi and Matsuoka examine cardiac energy dynamics during exercise loading by constructing and analyzing a sophisticated cellular model with excitation-contraction-energetics coupling. This study builds on previous work by this group that formulated a mitochondrial Ca^{2+} -dependent energy metabolism regulatory mechanism and recapitulated SR Ca^{2+} dynamics by considering Ca^{2+} interactions between mitochondria and the sarcoplasmic reticulum (SR) and proposes an extremely sophisticated new integrated human ventricular myocyte model that integrates the SR-mitochondrial Ca^{2+} dynamics with the human ventricular myocyte model (here based on the Grandi-Bers 2011 model). They quantified cardiac energy dynamics during dynamic load transitions during exercise and the contribution of Ca^{2+} -dependent regulatory mechanisms controlling mitochondrial energy metabolism to cardiomyocyte function.

The strengths of this study are the detailed analysis of changes in all factors of heart rate, preload, and contractility during dynamic load transitions during exercise, or the contribution to cardiac energy dynamics associated with these individual changes, and the systematic analysis of the effects of the presence or absence of Ca^{2+} -dependent energy metabolism regulatory mechanisms within mitochondria on cardiomyocyte function. In particular, an important role in the spatial interaction between mitochondrial Ca^{2+} uniporters and ryanodine receptors was identified, demonstrating that Ca^{2+} -dependent activation of NADH-producing dehydrogenases within mitochondria contributes to NADH stabilization during exercise. Such superb sophisticated and integrated models from the standpoint of systems physiology have the potential to establish a new assay system using in silico systems.

The manuscript is detailed and well-described. Hence, this reviewer will only point out a few errors in the manuscript and its presentation.

Thank you for your high evaluation on our work. We have revised the manuscript according to the comments.

Please see the following:

1. Please unify the abbreviations for mitochondrial Ca²⁺ uniporter; MCU and CaUni are mixed in the manuscript.

Thank you for the comment. We intended to use “MCU” to refer to it as a molecule, and “CaUni” to refer to it as an activity. However, these abbreviations were not sufficiently clear. We clarified the definitions in their first appearance underlined as follows;

“MCU complexed with accessory proteins such as MICUs, EMRE, and so on constituting CaUni activity, NCLX for NCX_{mit} activity, and Letm1 or TMBIM5 for HCX_{mit} activity” (p4, lines 5-7).

In addition, “Abstract” and “Key points summary” were revised underlined as follows;

“Abstract” “Simulation analyses revealed that the spatial coupling of mitochondria and SR, particularly via mitochondrial Ca²⁺ uniport activity-RyR,” (p3, lines 12-14).

“Key points summary 1” “Mitochondrial Ca²⁺ uniporter protein MCU and Na⁺-Ca²⁺ exchanger protein NCLX were reported to exist in proximity to sarcoplasmic reticulum (SR) ryanodine receptor RyR and Ca²⁺ pump SERCA, respectively, creating mitochondria-SR Ca²⁺ interaction in cardiomyocyte.” (p1)

“Key points summary 4” Simulation analyses revealed that the spatial coupling particularly via mitochondrial Ca²⁺ uniport activity-RyR is the primary determinant of Ca²⁺_{mit} concentration, and that the activation of NADH-producing dehydrogenases by Ca²⁺_{mit} contributes to NADH stability during exercise. (p2)

2. P.7, Lines 148-149: The mitochondria-SR interaction (MSI) component has been developed based on experimental observations using HL-1 cells (based on Takeuchi & Matsuoka, 2022). However, in integrating it into the human cardiomyocyte model, is there any part of the HL-1 cell model that has been adjusted to fit the human cardiomyocyte model? This should be mentioned if adjustments have been made based on in vitro experimental data using cardiomyocytes in humans.

Thank you for the comment. We developed mitochondria-SR interaction (MSI) component in the HL-1 model based on the spatial distribution of the molecules in mice ventricular myocytes obtained by super-resolution imaging, with equations for CaUni

and NCX_{mit} were adapted from our previous models of B lymphocytes, HL-1 cells, and isolated mitochondria (Kim et al., *J Physiol*, 2012; Takeuchi et al., *Sci Rep*, 2013; Saito et al., *J Physiol*, 2016). In integrating this MSI component into the human cardiomyocyte model, unfortunately, no quantitative data using cardiomyocytes in humans were available. However, we updated CaUni and NCX_{mit} models according to their electrophysiological characteristics as mentioned in the “Methods” section (p8, lines 7-22).

3. p.8, Lines 159-160: “100/70 times” This expression is different from the rest of the document, is there a reason for this?

Thank you for the comment. We rephrased the description to improve clarity underlined as follows;

“The amplitude factor for SERCA in the non-MSI model was set to 1.43 times, i.e., 100%/70%, that of the MSI model.” (p8, lines 5-6).

4. p.9, Line 199: I wonder if the result of Pi vs NADH_{mit} presented at the bottom of Appendix Fig. A1B does not reproduce the experimental data well. Any comments?

Thank you for the comment. Our previous model of isolated mitochondria (Saito et al., *J Physiol*, 2016) could perfectly reproduce the experimental data on Pi_{cyt} vs NADH_{mit} relationships, which were mainly caused by Pi_{mit} -dependent activation of Complex III. However, this regulation mechanism was subsequently denied by experimentally (Bazil et al., *Biophys J*, 2016; Vinnakota et al., *Biophys J*, 2016). Removing the mechanism resulted in the deviation from experimental data (Appendix Figure A1B). We compromise with the qualitatively comparable results; the increase of NADH_{mit} with Pi_{cyt} increase for ADP (–) condition and relatively small NADH_{mit} change with Pi_{cyt} increase for ADP (+) condition. We rephrased the description to improve clarity underlined as follows;

“The updated isolated mitochondrial model well reproduced the *in vitro* experiments on cytoplasmic $\text{Ca}^{2+}_{\text{cyt}}$ -dependent changes of NADH_{mit} and mVO_2 using isolated cardiac mitochondria (Appendix Figure A1A) (Territo et al., 2000). In addition, qualitatively comparable results were obtained for cytoplasmic Pi (Pi_{cyt})-dependency observed using isolated cardiac mitochondria (Bose et al., 2003) (Appendix Figure A1B), albeit with deviations due to the removal of Pi_{mit} -dependent activation of Complex III.” (p10, lines 3-8).

We also corrected x-axis labels in Appendix Figure A1, by removing “[]”, to unify the presentation throughout the all Figures.

5. p.12, Line 254: given the statement in p.11, Lines 239-240, is "0.3" a mistake for 0.35? And then $P_{Ca,0}$, $P_{Na,0}$, and $PK,0$ need to be explained.

Thank you for pointing this out. The increasing extent of the amplitude factors for L-type Ca^{2+} current (I_{CaL}) and SERCA, "1.35 times", was a mistake for "1.30 times". Therefore, the formula (p12 line 254 in the original version) was correct, but the description (p11, line 240 in the original version) was wrong. In addition, the F_{ext} increase under the standard exercise condition, "4.25" (p11, line 237 in the original version) was a mistake for "5.00". We apologize for these mistakes and fixed them (p11, lines 20 and 23).

We explained the definitions of $P_{Ca,0}$, $P_{Na,0}$, $P_{K,0}$ and also of PCa , PNa , and PK , in the text as follows;

"where PCa , PNa , and PK are permeability factors for Ca^{2+} , Na^+ , and K^+ , respectively, and $P_{Ca,0}$, $P_{Na,0}$, and $P_{K,0}$ are permeability factors in the absence of β -adrenergic receptor stimulation for Ca^{2+} , Na^+ , and K^+ , respectively." (p12, lines 17-19).

6. P.12, Line 255: Similar to #5, "0.3" could be a mistake for 0.35. Further explanation of $V_{max_SERCA,0}$ is also needed.

Thank you for pointing this out. As mentioned above, the increasing extent of the amplitude factors for L-type Ca^{2+} current (I_{CaL}) and SERCA, "1.35 times", was a mistake for "1.30 times". We apologize for this mistake and fixed it (p11, line 23).

We explained the definitions of V_{max_SERCA} and $V_{max_SERCA,0}$ in the text as follows;
"where V_{max_SERCA} is maximum velocity and $V_{max_SERCA,0}$ is maximum velocity in the absence of β -adrenergic receptor stimulation." (p12, lines 23-24).

7. P.12, Line 256: $V_{max_INaK,0}$ explanation needed.

We explained the definitions of V_{max_INaK} and $V_{max_INaK,0}$ in the text as follows;
"where V_{max_INaK} is maximum velocity and $V_{max_INaK,0}$ is maximum velocity in the absence of β -adrenergic receptor stimulation." (p13, lines 3-4).

8. P.12, Line 257: Explanation of $G_{Ks,0}$ needed.

We explained the definitions of G_{Ks} , $G_{Ks,0}$, V_{shift} , $x_{Ks_{ss}}$, and $t_{x_{Ks}}$ in the text as follows;
"where G_{Ks} , $G_{Ks,0}$, V_{shift} , $x_{Ks_{ss}}$, and $\tau_{x_{Ks}}$ are conductance, conductance in the absence of β -adrenergic receptor stimulation, voltage shift factor in the presence of β -adrenergic

receptor stimulation, steady-state value of gate, and time constant of gate, respectively.” (p13, lines 11-13).

Accordingly, the subsequent sentence was rephrased as follows;

“For the β -adrenergic receptor stimulation, α value is varied 0–3.0.” (p13, line 14).

9. P.13, Lines 264-266: Shouldn't authors comment on the correspondence with the original formula (supplementary materials)?

Thank you for pointing this out. We explained the definitions of parameters for PDHC, ICDH and OGDH formula, and commented on the correspondence with the original formula in the Online Supplementary Material as follows;

“where f_{PDHa} , K_{Ca} , and n_{Ca} are activation factor by Ca^{2+}_{mit} , binding constant of Ca^{2+}_{mit} , and Hill coefficient, respectively. u_1 and u_2 are model fitted factors.” (p14, lines 1-2).

“where Ca_{ACT} , K_{Ca} , and K_{iMg2} are activation factors by Ca^{2+}_{mit} , binding constants of Ca^{2+}_{mit} , and binding constant for inhibition by Mg^{2+}_{mit} , respectively. α and β are model fitted factors.” (p14, lines 10-11).

“where Ca_{ACT} , K_{Ca} , and n_{Ca} are activation factors by Ca^{2+}_{mit} , binding constant of Ca^{2+}_{mit} , and Hill coefficient, respectively. α and β are model fitted factors.” (p14, lines 16-17).

“For the control simulations with allosteric Ca^{2+}_{mit} -dependent regulations, Ca^{2+}_{mit} was used instead of FCa_{mit} (Saito et al., 2016). Full equations are summarized in the Online Supplementary Material Table S9.” (p14, last line to p15, lines 1-2).

10. Essentially, an explanation of each symbol in equations on p. 12 and p. 13 is needed. It may be complicated.

Thank you for the comments. We explained each symbol in equations (p12 and p13 in the original version) as described in the answers to questions #5-#9. In addition, explanation of each symbol in equations for Ca_{uni} and NCX_{mit} were also added underlined as follows;

“where apparent inhibition constant of Ca^{2+}_{mit} , $K_{inh} = 5.0 \times 10^{-5}$ mM, apparent recovery constant of Ca^{2+}_{mit} , $K_{rec} = 8.0 \times 10^{-4}$ mM, Hill coefficient $n = 2$.” (p8, lines 10-11)

“The NCX_{mit} model (Takeuchi & Matsuoka, 2022) was updated by increasing the Ca^{2+}_{mit} affinity; i.e., reducing the apparent binding constant of Ca^{2+}_{mit} , $Kd_{Ca_{mit}}$, from 0.0209 to 0.0025 mM, based on our electrophysiological measurements using mouse cardiac mitoplast (Islam et al., 2020). Accordingly, apparent binding constant of mitochondrial Na^+ , $Kd_{Na_{mit}}$, was reduced from 38.0000 to 18.7137 mM to satisfy the constraint of the equilibrium potential of NCX_{mit} , $E_{Na/Ca}$ (Kim & Matsuoka, 2008);” (p8, lines 12-17).

“where E_{Na} , E_{Ca} , $KdNa_{cyt}$, and $KdCa_{cyt}$ are equilibrium potential of Na^+ , equilibrium potential of Ca^{2+} , apparent binding constant of cytosolic Na^+ (Na^+_{cyt}), and apparent binding constant of Ca^{2+}_{cyt} , respectively.” (p8, lines 20-22).

11. P.15, Lines 318-319: *Looking at the left panel of Figures 6c and 6d, there is indeed an increase in flux. However, although it is stated in the main text that both consumption and production fluxes are increasing, they do not appear to be depicted together. Thus, the flux difference plots do not reveal what the difference is.*

Thank you for pointing this out. The legends for Figure 6C and D were missing, which made the referee difficult to read data. We apologize for this mistake. Indeed, consumption (solid lines) and production (dashed lines) fluxes were almost overlapped in the left panels of Figure 6C and D, but small difference did exist as shown in the right panels of Figure 6C and D. We added the notes in the Figure 6C and D, in addition to revising Figure legends as follows;

“**C** and **D**. NADH-consuming flux of Complex I (solid lines) and NADH-producing fluxes of all dehydrogenases, PDHC, ICDH, OGDH, and MDH (dashed lines) (left panels), and the difference, NADH-producing flux minus NADH-consuming fluxes (right panels) in the simulations shown in **A** and **B**, respectively.” (p46).

12. Fig.6 Legend: *it lacks a brief description of panels C and D.*

Thank you for pointing this out. As mentioned in the answer to question #11, we corrected the legend of Figure 6C and D (p46). We apologize for this mistake.

13. P.17, Lines 363-364: *I couldn't understand where this sentence refers to. Does the flux in Complex I refer to Fig. 8A?*

Thank you for pointing this out. As the referee assumes, the flux in Complex I refers to Figure 8A. We added “(see Figure 8A)” in the sentence to improve clarity (p18, last two lines).

14. P.18, Line 372: *“0.83-0.33 sec”; On the other hand, the figure legend of Fig. 9 says the CL shortening of 1.0-0.33 sec. Which is correct? In addition, shouldn't the CL shortening cases be clearly stated? For example, 1.0, 0.83, 0.5, 0.33, etc.*

Thank you for pointing this out and for helpful comments. The cycle length was varied with 1.00-0.33 sec. We rephrased the description to improve clarity underlined as follows;

“The cycle length shortenings from 1.00 to 1.00–0.33 sec with (green closed circles)” (p19, line 8).

In addition, we stated clearly the simulation conditions for cycle length shortenings, F_{ext} increases, and α value for β -adrenergic stimulation in the Figure 9 legend underlined as follows;

“cycle length (CL) shortening from 1.00 to 1.00, 0.83, 0.67, 0.59, 0.50, 0.40, 0.33 sec.....” (p47).

“ F_{ext} increase from 0.00 to 0.00, 0.90, 1.98, 3.34, 5.00, 7.25, and 10.00.....” (p47).

“application of $\alpha=0.0, 0.5, 1.0, 1.5, 2.0, 2.5, 3.0$ for β -adrenergic stimulation.....” (p47).

15. P.18, Line 388: "Ca_{2+mi}" Typo? Correct "Ca_{2+mit}"?

Thank you for pointing this out. We corrected the typo “Ca_{2+mi}” to “Ca_{2+mit}” (p19, line 24).

16. 32. p.19, Lines 411-412: "However, ... diminished." Where should we look for the results in this sentence? The Ca_{2+cyt} transient, the decrease in Ca_{2+mit}, and the decrease in Ca₂₊ flux via RyR can be understood from Fig. A5 (but I think it should be clearly stated to be compared to Fig. 2A). On the other hand, where should we look for hsmL shortening and decreased ATP consumption.

Thank you for pointing this out and helpful comments. We introduced subpanel labels in the Appendix Figures A2 and A5 and cited in the manuscript. In addition, we added “E. Contraction” section composed of “a. hsmL (μm)” and “b. Force (mN/mm^2)” in these figures. Accordingly, the descriptions were updated underlined as follows; “Action potential configuration and most ionic currents were not largely affected (compare Appendix Figure A2A and A5A). However, Ca_{2+cyt} transients, Ca_{2+mit}, and therefore the extents of active force generation and hsmL shortening, as well as Ca₂₊ flux via RyR were diminished (see Appendix Figure A5Da, De, Eb, Ea, and Ba), resulting in lower ATP consumption (Table 1).” (p20, line 23 to p21, line 2).

17. p.19, Lines 413-414: "Accordingly, ... MSI model." Where is this sentence attributed from?

This sentence is attributed from Table 1. We added “(Table 1)” at the end of the sentence to improve clarity (p21, line 3).

18. P.22, Line 493: "CaUni_t"; is this typo? CaUni_{cyt} or CaUni?

Thank you for pointing this out. We corrected the typo “CaUni_t” to “CaUni” (p24, line 7).

19. P.57, Figure 9: *The plot is a little difficult to read, so please revise it.*

Thank you for the comment. To help readers easily compare “Control” and “Silenced Ca²⁺-regulation”, we enlarged the symbols as well as the panels, and added grids to the panels in Figure 9. In addition, according to the question by the referee #1, we added new panels for the heart rate-PCr/ATP relationships (Figure 9D and H).

Dear Dr Takeuchi,

Re: JP-RP-2024-286054R1 "A simulation study on the role of mitochondria-sarcoplasmic reticulum Ca²⁺ interaction in cardiomyocyte energetics during exercise" by Ayako Takeuchi and Satoshi Matsuoka

We are pleased to tell you that your paper has been accepted for publication in The Journal of Physiology.

Authors should note that it is too late at this point to offer corrections prior to proofing. Major corrections at proof stage, such as changes to figures, will be referred to the Editors for approval before they can be incorporated. Only minor changes, such as to style and consistency, should be made at proof stage. Changes that need to be made after proof stage will usually require a formal correction notice.

If you would like to receive our 'Research Roundup', a monthly newsletter highlighting the cutting-edge research published in The Physiological Society's family of journals (The Journal of Physiology, Experimental Physiology and Physiological Reports), please click this link, fill in your name and email address and select 'Research Roundup':
<https://www.physoc.org/journals-and-media/membernews/>.

Yours sincerely,

Vaughan Macefield
Senior Editor
The Journal of Physiology

P.S. - You can help your research get the attention it deserves! Check out Wiley's free Promotion Guide for best-practice recommendations for promoting your work at www.wileyauthors.com/eeo/guide. You can learn more about Wiley Editing Services which offers professional video, design, and writing services to create shareable video abstracts, infographics, conference posters, lay summaries, and research news stories for your research at www.wileyauthors.com/eeo/promotion.

IMPORTANT NOTICE ABOUT OPEN ACCESS: To assist authors whose funding agencies mandate public access to published research findings sooner than 12 months after publication, The Journal of Physiology allows authors to pay an Open Access (OA) fee to have their papers made freely available immediately on publication.

You can check if your funder or institution has a Wiley Open Access Account here: <https://authorservices.wiley.com/author-resources/Journal-Authors/licensing-and-open-access/open-access/author-compliance-tool.html>.

EDITOR COMMENTS

Reviewing Editor:

All point-by-point responses to the comments are truly thorough and relevant revisions are satisfactory. Both reviewers and I have no more comments and evaluate the scientific merits of this paper as highly influential. I would like to recommend acceptance in the present form.

I feel pleased to have this excellent commissioned paper in the Special Issue and would like to thank the authors for their contribution.

Senior Editor:

Thank you for submitting your revised manuscript to The journal of Physiology. As you will see, the Reviewing Editor and two independent reviewers are satisfied with your amendments and therefore I am pleased to report that your manuscript has been accepted for publication.

REFeree COMMENTS

Referee #1:

The authors have addressed all of my questions and concerns.

Referee #2:

The authors have nicely addressed all of my concerns and comments.

1st Confidential Review

08-Aug-2024